# Purkinje cell misfiring generates high-amplitude action tremors that are corrected by cerebellar deep brain stimulation

Amanda M Brown[1,2,3], Joshua J White[1,2,3], Meike E van der Heijden[1,3], Joy Zhou[1,2,3], Tao Lin[1,3], Roy V Sillitoe[1,2,3,4]*

[1]Department of Pathology and Immunology, Baylor College of Medicine, Houston, United States; [2]Department of Neuroscience, Baylor College of Medicine, Houston, United States; [3]Jan and Dan Duncan Neurological Research Institute of Texas Children's Hospital, Houston, United States; [4]Development, Disease Models & Therapeutics Graduate Program, Baylor College of Medicine, Houston, United States

**Abstract** Tremor is currently ranked as the most common movement disorder. The brain regions and neural signals that initiate the debilitating shakiness of different body parts remain unclear. Here, we found that genetically silencing cerebellar Purkinje cell output blocked tremor in mice that were given the tremorgenic drug harmaline. We show in awake behaving mice that the onset of tremor is coincident with rhythmic Purkinje cell firing, which alters the activity of their target cerebellar nuclei cells. We mimic the tremorgenic action of the drug with optogenetics and present evidence that highly patterned Purkinje cell activity drives a powerful tremor in otherwise normal mice. Modulating the altered activity with deep brain stimulation directed to the Purkinje cell output in the cerebellar nuclei reduced tremor in freely moving mice. Together, the data implicate Purkinje cell connectivity as a neural substrate for tremor and a gateway for signals that mediate the disease.

*For correspondence:
sillitoe@bcm.edu

Competing interests: The authors declare that no competing interests exist.

## Introduction

Tremors are uncontrollable muscle oscillations that result in rhythmic shaking of the affected body parts. Tremor occurs in healthy individuals at baseline, which is known as physiological tremor (*Raethjen et al., 2000*). However, tremor also occurs as a movement disorder when its amplitude becomes severe enough to disrupt daily activities (*Elias and Shah, 2014*). Tremor can constitute independent diseases, such as in essential tremor, the most common tremor disease (*Clark and Louis, 2018*; *Haubenberger and Hallett, 2018*). It can also be co-morbid with other brain disorders such as Parkinson's disease (*Hallett, 2014*), dystonia (*Defazio et al., 2015*; *Pandey and Sarma, 2016*), ataxia (*Hagerman and Hagerman, 2015*), or epilepsy (*Striano and Zara, 2016*). Additionally, tremor can be a negative consequence of a growing list of common prescription drugs, toxins, or neurological insults (*Bhatia et al., 2018*; *Morgan et al., 2017*). While there are a great number of diseases, disorders, and chemicals that are associated with tremor, the neural origins of tremor are largely not understood and they are especially unclear in the most common tremor diseases (*Hallett, 2014*; *Pedrosa et al., 2014*).

There is good evidence implicating dysfunctional cerebello-thalamo-cortical circuits in tremor. The cerebellar receiving areas of the thalamus such as the ventral intermediate nucleus (VIM) and the ventral anterolateral nucleus (VAL) are preferred targets for thalamotomy and deep brain

stimulation (DBS) in the treatment of essential tremor (*Pahwa et al., 2001*). Local field potentials and spike activity recorded from these brain areas in humans experiencing bouts of tremor correlate with the frequency of oscillation in the affected body parts (*Pedrosa et al., 2014*; *Hua et al., 1998*). However, it is unclear where in the brain this abnormal activity originates (*Pedrosa et al., 2014*). Functional magnetic resonance imaging (fMRI) studies in humans with essential tremor reported abnormal levels of activity in the cerebellum (*Broersma et al., 2016*). Compellingly, when brain activity of individuals with tremor disorders is compared between periods of mimed tremor and true epochs of tremor, the only area of the brain with significantly different patterns of activity is the cerebellum (*Bucher et al., 1997*). Yet, if and how the cerebellum could provide a major contribution to either generating the tremor – therefore, acting as an origin of the signal – or mediating the transfer of an existing tremor signal, has not been elucidated. Further, the respective role that individual cerebellar cell types may have in vivo in the behaving animal during the production or propagation of tremor-related neural activity is unknown.

Abnormalities in different cerebellar cell types, particularly the Purkinje cells, have been associated with tremor. However, it is unclear whether these neurons are directly responsible for generating or propagating tremor (*Louis, 2016*). Likely, the positioning of each cell type within the local cerebellar circuitry, as well as the motor circuit as a whole, guides how each one influences tremor (*Ito, 2006*). The cerebellar cortex has a canonical and repeating architecture throughout all of its lobules and is comprised of the Purkinje cells at the center of a microcircuit that integrates information from five major classes of excitatory and inhibitory interneurons. Inputs to the cerebellum include mossy fibers and climbing fibers, with the latter originating in the inferior olive where it sends powerful excitatory inputs directly onto the Purkinje cell dendrites. Purkinje cells project out of the cerebellar cortex to make inhibitory synapses onto the cerebellar nuclei neurons. The cerebellar nuclei provide the final output of the cerebellum, representing the culmination of all cerebellar inputs and computations therein. Therefore, current views consider the cerebellar nuclei signals as a link between the cerebellum and the rest of the brain and spinal cord and Purkinje cell activity as the computational center that shapes these signals (*D'Angelo, 2018*; *Figure 1a–c*).

Accordingly, there is compelling, albeit indirect, evidence pointing to a role for abnormal, reduced, or the loss of Purkinje cell to cerebellar nuclei communication as a key neural substrate for tremor (*Handforth, 2016*). Additionally, studies have found abnormalities in cells directly upstream (*Erickson-Davis et al., 2010*) as well as in the cerebellar nuclei directly downstream (*Paris-Robidas et al., 2012*) of Purkinje cells to be associated with tremor. While Purkinje cell loss and degenerative Purkinje cell morphology have been noted in some types of tremor, these hallmarks are not found across all tremor conditions (*Morgan et al., 2017*; *White et al., 2016a*). It follows that, with the many varied potential causes and diseases associated with tremor, there may be equally as many potential biological substrates of tremor. In the face of this problem, we have sought to determine whether cerebellar Purkinje cells have a direct role in tremor generation and, if so, whether there are electrophysiological abnormalities in the cerebellum that dictate the tremor state.

## Results

### Lack of Purkinje cell GABA neurotransmission does not induce pathological tremor

Previous human pathology studies of essential tremor raised the possibility that loss or reduction of Purkinje cell signaling causes tremor (*Paris-Robidas et al., 2012*; *Axelrad et al., 2008*). In order to address whether Purkinje cells have a role in the production and propagation of tremor signals, we first tested whether removing Purkinje cell to cerebellar nuclei neurotransmission triggers tremor. To accomplish this, we used a $Pcp2^{Cre}$;$Slc32a1^{flox/flox}$ conditional genetic approach to delete the vesicular GABA transporter (VGAT) from Purkinje cells (*Lewis et al., 2004*; *Tong et al., 2008*). The result of this manipulation is that Purkinje cells can still fire simple spike and complex spike action potentials, but they can no longer communicate with their downstream partners using fast GABA neurotransmission, which ultimately results in silencing the Purkinje cell output (*Figure 1d*). Using anesthetized mice, we previously demonstrated that genetic deletion of $Slc32a1$, the gene encoding VGAT, in Purkinje cells results in alterations in Purkinje cell firing activity with consequent changes in

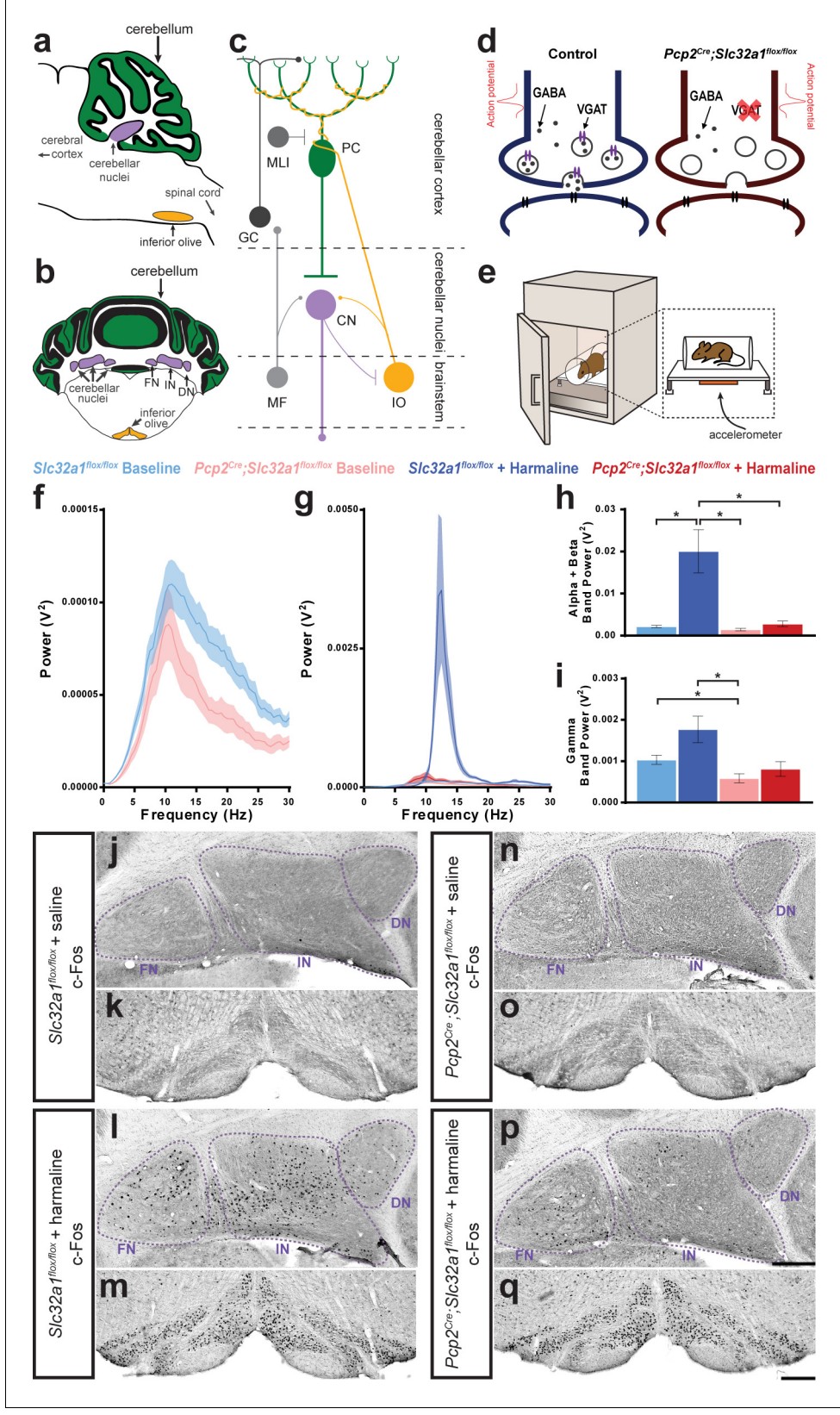

**Figure 1.** Purkinje cell neurotransmission is necessary for producing baseline physiological tremor and pathological tremor. (a) Representation of a sagittal section of a mouse cerebellum indicating its spatial relationship to other landmarks of the central nervous system. Green region = Purkinje cell dendrites and

*Figure 1 continued on next page*

*Figure 1 continued*

molecular layer, purple region = cerebellar nuclei, yellow region = inferior olive. (**b**) Representation of a coronal section through a mouse cerebellum where the fastigial nucleus (FN), interposed nucleus (IN), and dentate nucleus (DN) are all visible. Green region = Purkinje cell dendrites and molecular layer, purple regions = cerebellar nuclei, yellow regions = inferior olive. (**c**) Representation of a simplified cerebellar circuit including a Purkinje cell (PC, green), cerebellar nuclei (CN, purple), inferior olive (IO, yellow), mossy fibers (MF), granule cell (GC), and molecular layer interneuron (MLI). Large circles = cell bodies, small circle terminals = excitatory synapses, flat terminals = inhibitory synapses. (**d**) Representation of the result of genetic manipulation in *Pcp2^Cre^;Slc32a1^flox/flox^* mice. Control Purkinje cell synapse depicted in blue on left, *Pcp2^Cre^;Slc32a1^flox/flox^* Purkinje cell synapse depicted in red on right. Large open circles = vesicles. Small filled circles = GABA. Purple ellipse pairs = VGAT. Bright red action potential cartoon represents an action potential reaching the synapse and triggering the fusion of vesicles to the presynaptic membrane and release of the vesicles' contents, such as GABA, onto receptors in the postsynaptic membrane (black ellipse pairs). GABA is released from Purkinje cells during fast neurotransmission in *Slc32a1^flox/flox^* mice, but not in *Pcp2^Cre^;Slc32a1^flox/flox^* mice. (**e**) Representation of a commercial tremor monitor. Inset = dotted rectangle. Accelerometer = orange rectangle. (**f–g**) Solid line = mean. Shaded region = standard error of the mean (SEM). Legend above. Source data available in *Figure 1—source data 1*. (**f**) Mice lacking Purkinje cell GABA neurotransmission had lower baseline physiological tremor compared to control animals. Control N = 16, mutant N = 12. (**g**) While control animals exhibited the typical robust tremor after harmaline administration (N = 16), *Pcp2^Cre^;Slc32a1^flox/flox^* animals had no significant increase in tremor in response to the drug (N = 13). The baseline data from **f** are repeated on this graph for scale. (**h**) Summed tremor power within the alpha and beta bands. Legend above. (**i**) Summed tremor power within the gamma band. Legend above. Source data for **h** and **i** are available in *Figure 1—source data 1*. (**j–q**) c-Fos expression in the cerebellar nuclei (**j, l, n, p**) and inferior olive (**k, m, o, q**) after saline (**j–k, n–o**) or harmaline (**l–m, p–q**) administration. For the tremor recordings, we define baseline as it relates to the conditions performed with and without harmaline, whereas the saline injection group relates to the experiments in which c-Fos measurements were carried out. Cerebellar nuclei scale = 250 µm. Inferior olive scale = 250 µm.

The online version of this article includes the following source data and figure supplement(s) for figure 1:

**Source data 1.** Source data for representative graphs in *Figure 1*.
**Figure supplement 1.** Baseline tremor power of both genotypes and power of tremor in recordings of *Pcp2^Cre^; Slc32a1^flox/flox^* mice after harmaline administration are an order of magnitude smaller than that of *Slc32a1^flox/flox^* mice after harmaline administration.
**Figure supplement 1—source data 1.** Precision measures, exact p-values, and replicate data relevant to *Figure 1*.
**Figure supplement 2.** No difference in tremor was found between males and females.
**Figure supplement 2—source data 1.** Source data for representative graphs in *Figure 1—figure supplement 2*.
**Figure supplement 3.** Mice lacking Purkinje cell GABA neurotransmission have reduced c-Fos expression in response to harmaline administration in the cerebellar nuclei, despite similar levels of activation in the inferior olive.
**Figure supplement 3—source data 1.** Source data for representative graphs in *Figure 1—figure supplement 3*.

cerebellar nuclei neuron firing frequency and regularity (*White et al., 2014*). The somewhat paradoxical effect that Purkinje cells have on their own firing activity in the mutants likely arises, at least in part, because Purkinje cells project to the cerebellar nuclei, which project to the inferior olivary nucleus in the brainstem, which then projects back to Purkinje cells to form a tri-synaptic closed loop circuit (*White et al., 2014*; *Chaumont et al., 2013*; *Witter et al., 2013*). The anatomical fidelity of the cerebellar circuit is maintained despite this manipulation of neuronal firing properties, though ataxia is present (*White et al., 2014*; *Video 1*).

A possible additional behavioral outcome in mice with silenced Purkinje cell output would be a tremor phenotype. However, we found that mice without Purkinje cell output (*Pcp2^Cre^; Slc32a1^flox/flox^*) did not have an enhanced tremor

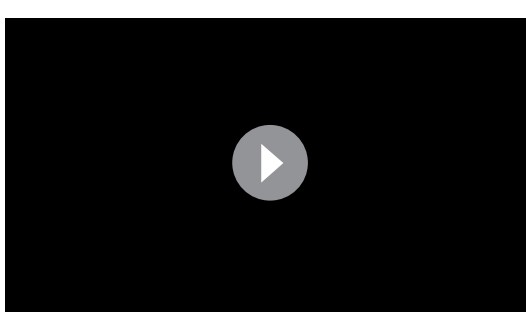

**Video 1.** *Pcp2^Cre^;Slc32a1^flox/flox^* mice exhibit ataxia and disequilibrium, but not pathological tremor.
https://elifesciences.org/articles/51928#video1

phenotype compared to control mice ($Slc32a1^{flox/flox}$). Instead, the lack of Purkinje cell GABA neurotransmission resulted in a lower than normal baseline of physiological tremor (*Figure 1e–f and i*, *Figure 1—figure supplement 1*, *Table 1*, $Slc32a1^{flox/flox}$ (referred to from here on as control) *baseline N = 16*, $Pcp2^{Cre};Slc32a1^{flox/flox}$ (referred to from here on as mutant) *baseline N = 12*). These data suggest that Purkinje cell activity may have a role in establishing the normal level of physiological tremor but the lack of Purkinje cell output activity alone is not a sufficient functional change in the cerebellar circuit to result in pathological levels of tremor. Importantly, the reduction in baseline physiological tremor was observed only in the gamma band. Changes in gamma band frequency reflect alterations in motor behavior after loss of Purkinje cell neurotransmission (*Figure 1i*). The sex of the mice did not affect tremor at baseline (*Figure 1—figure supplement 2*). The resistance to change in alpha/beta band frequencies raises an interesting problem about whether frequencies that are common to tremor diseases depend on Purkinje cell function.

## Lack of Purkinje cell neurotransmission reduces harmaline tremor

Since the lack of Purkinje cell neurotransmission did not produce tremor, we next sought to determine the role of Purkinje cells in the context of a potentially greater tremor circuit. For this, we administered harmaline, a beta-carboline alkaloid compound that causes an 11–14 Hz tremor in mice and 8–16 Hz tremor in multiple species, including humans (*Handforth, 2012*). Harmaline affects many types of receptors, ion channels, and gap junctions that are found throughout the nervous system, and therefore likely affects the activity of multiple cell populations in the brain (*Handforth, 2015*). However, a rich history of research in slice (*Park et al., 2010*), decerebrate (*Lamarre et al., 1971*; *Llinás and Volkind, 1973*), and anesthetized (*Park et al., 2010*; *Llinás and Volkind, 1973*) preparations have indicated that harmaline-induced tremor involves synchronous rhythmic firing in the inferior olive. If this is the case, then one would postulate that Purkinje cells must be involved in the production of the tremor response. The influence of Purkinje cells during this process has remained unclear due to complicated results in genetic approaches, diverse circuit manipulation techniques, a confounding lack of cell type specificity, or the presence of degeneration

**Table 1.** Precision measures, exact p-values, and replicate data relevant to *Figure 1*.

| Figure | Comparator 1 | Comparator 2 | Mean 1 | Mean 2 | SE of diff. | N 1 | N 2 | Summary | Adjusted P Value |
|---|---|---|---|---|---|---|---|---|---|
| *Figure 1h* | $Slc32a1^{flox/flox}$ baseline | $Slc32a1^{flox/flox}$ + harmaline | 0.002181 $V^2$ | 0.02005 $V^2$ | 0.005132 | 16 | 16 | * | 0.0190 |
| | $Slc32a1^{flox/flox}$ baseline | $Pcp2^{Cre};Slc32a1^{flox/flox}$ baseline | 0.002181 $V^2$ | 0.001438 $V^2$ | 0.0003786 | 16 | 12 | ns | 0.2982 |
| | $Slc32a1^{flox/flox}$ baseline | $Pcp2^{Cre};Slc32a1^{flox/flox}$ + harmaline | 0.002181 $V^2$ | 0.002787 $V^2$ | 0.0007072 | 16 | 13 | ns | 0.9420 |
| | $Slc32a1^{flox/flox}$ + harmaline | $Pcp2^{Cre};Slc32a1^{flox/flox}$ baseline | 0.02005 $V^2$ | 0.001438 $V^2$ | 0.005133 | 16 | 12 | * | 0.0142 |
| | $Slc32a1^{flox/flox}$ + harmaline | $Pcp2^{Cre};Slc32a1^{flox/flox}$ + harmaline | 0.02005 $V^2$ | 0.002787 $V^2$ | 0.005168 | 16 | 13 | * | 0.0252 |
| | $Pcp2^{Cre};Slc32a1^{flox/flox}$ baseline | $Pcp2^{Cre};Slc32a1^{flox/flox}$ + harmaline | 0.001438 $V^2$ | 0.002787 $V^2$ | 0.0007151 | 12 | 13 | ns | 0.3550 |
| *Figure 1i* | $Slc32a1^{flox/flox}$ baseline | $Slc32a1^{flox/flox}$ + harmaline | 0.001032 $V^2$ | 0.001767 $V^2$ | 0.0003403 | 16 | 16 | ns | 0.2224 |
| | $Slc32a1^{flox/flox}$ baseline | $Pcp2^{Cre};Slc32a1^{flox/flox}$ baseline | 0.001032 $V^2$ | 0.0005844 $V^2$ | 0.0001552 | 16 | 12 | * | 0.0454 |
| | $Slc32a1^{flox/flox}$ baseline | $Pcp2^{Cre};Slc32a1^{flox/flox}$ + harmaline | 0.001032 $V^2$ | 0.0008114 $V^2$ | 0.0002104 | 16 | 13 | ns | 0.8693 |
| | $Slc32a1^{flox/flox}$ + harmaline | $Pcp2^{Cre};Slc32a1^{flox/flox}$ baseline | 0.001767 $V^2$ | 0.0005844 $V^2$ | 0.0003403 | 16 | 12 | * | 0.0155 |
| | $Slc32a1^{flox/flox}$ + harmaline | $Pcp2^{Cre};Slc32a1^{flox/flox}$ + harmaline | 0.001767 $V^2$ | 0.0008114 $V^2$ | 0.0003688 | 16 | 13 | ns | 0.0897 |
| | $Pcp2^{Cre};Slc32a1^{flox/flox}$ baseline | $Pcp2^{Cre};Slc32a1^{flox/flox}$ + harmaline | 0.0005844 $V^2$ | 0.0008114 $V^2$ | 0.0002104 | 12 | 13 | ns | 0.8543 |

that interferes with interpretation of neuronal function (*Llinás and Volkind, 1973*; *Milner et al., 1995*; *Kralic et al., 2002*).

We found that mice lacking Purkinje cell signaling did not have a significantly increased tremor after harmaline administration compared to control genotype animals, which displayed the typical robust tremor in response to the drug (*Figure 1e and g–h*, *Figure 1—figure supplement 1*, *Table 1*, *Video 2*. Control + harmaline N = 16, mutant + harmaline N = 13). The strong harmaline tremor peak in control animals spanned the alpha and beta bands. This represented a shift in peak tremor frequency for control animals from a baseline peak at 11 Hz to a harmaline tremor peak at 12.5 Hz. Meanwhile, little shift in tremor peak was noted for the mutant mice for which baseline peak is at 10.5 Hz, and after harmaline administration is at 10 Hz (*Figure 1—figure supplement 1*). The slight difference between peak frequencies when comparing both mutant conditions, treated and untreated, to control baseline peak is likely due to movement specific differences between the two genotypes. When observing the power of frequencies in the gamma band, away from the peak harmaline frequencies (*Figure 1i*), we find that the control + harmaline and mutant + harmaline conditions are not significantly different from one another. However, we note that control + harmaline tremor is still elevated at these frequencies and hypothesize this is because of variation in the motor responses of the animals, specifically at gamma frequencies that are involved in movement (*van Wijk et al., 2012*; *Armstrong et al., 2018*). This is pronounced in the control + harmaline condition because of the severe tremor that occurs during different movements. This is of particular relevance since our tremor recordings are conducted during freely-moving behavior when the mice are able to explore in a five inches (length) by 4.5 inches (width) box arena, which allows the elucidation of expected moment-to-moment and mouse-to-mouse variations. The sex of the mice did not affect the reliability of inducing a strong harmaline tremor (*Figure 1—figure supplement 2*). Taking these results together, the resistance to change in alpha/beta band frequencies after deleting VGAT in Purkinje cells, particularly in the presence of harmaline, supports the hypothesis that particular frequencies that are common to tremor diseases depend on proper Purkinje cell function (*Figure 1h*).

We next asked how key nodes in the cerebellar system collectively respond to changes in circuit activity after harmaline is provided by quantifying the activation of the early activation transcription factor c-Fos. While there was little to no activation of c-Fos in saline-treated animals of both genotypes (*Figure 1j–k and n–o*), harmaline administration resulted in robust activation of c-Fos in both the cerebellar nuclei and inferior olive of control genotype animals, as expected (*Figure 1j–m*; *Tian and Bishop, 2002*; *Oldenbeuving et al., 1999*; *Miwa et al., 2000*; *Beitz and Saxon, 2004*). However, compared to control mice, the extent of cerebellar nuclei c-Fos activation in response to harmaline was significantly reduced in mice with silenced Purkinje cell output (*Figure 1l and p*). This is despite having similar activation of the inferior olive (*Figure 1m and q*). When quantified, this observation was maintained for all three nuclei with the largest magnitude difference between mice with and without Purkinje cell GABA neurotransmission occurring in the interposed nucleus in terms of density of c-Fos expressing cells (*Figure 1—figure supplement 3a* and *Figure 1—figure supplement 3—source data 1*). This finding was supported by measurements of the percent of the area of the cerebellar nuclei covered by c-Fos activation, a representation of the robustness of the c-Fos activation within cells (*Figure 1—figure supplement 3b* and *Figure 1—figure supplement 3—source data 1*). This change occurred despite similar levels of activation of the inferior olive in both genotypes (*Figure 1—figure supplement 3c-d* and *Figure 1—figure supplement 3—source data 1*). Compared to the fastigial and interposed, the dentate nucleus has overall lower c-Fos expression after harmaline treatment, which is consistent with previous reports (*Beitz and Saxon, 2004*; *Oldenbeuving et al., 1999*). It is also noted that, while there is significantly less activation of c-Fos in the mutant mice compared to control animals after harmaline administration, there is still some level of activation of the cerebellar nuclei despite the silencing

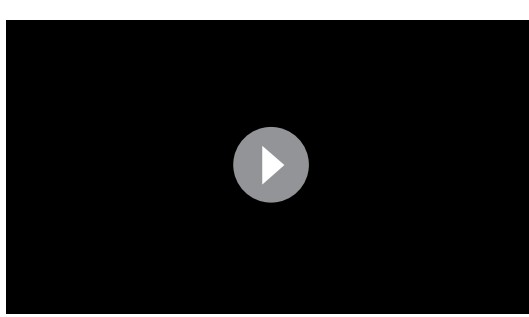

**Video 2.** Harmaline induces a severe tremor in *Slc32a1^{flox/flox}* mice but not in *Pcp2^{Cre};Slc32a1^{flox/flox}* mice.
https://elifesciences.org/articles/51928#video2

of Purkinje cell GABA neurotransmission. One possible cause for this activation are the collaterals from the inferior olive that project to the cerebellar nuclei (*Figure 1c*). The effect of these collaterals on the cerebellar nuclei gradually reduces over the course of development (*Najac and Raman, 2017*). Typically, by adulthood these collaterals have little effect on cerebellar nuclei activity, and any excitation is usually masked by the strong inhibitory input imparted by the Purkinje cells (*Lu et al., 2016*), but it is possible that harmaline's abnormally strong activation of the inferior olive may be capable of driving these collaterals to produce an excitatory effect on the nuclei to induce some residual c-Fos activation. Interestingly, as previously described (*White et al., 2014*) and also discussed below (*Figure 2*; *Table 2*; *Figure 3*; *Table 3*) the mutant mice have abnormal cerebellar neuron activity. However, the presence of c-Fos does not represent long-term differences in baseline neuronal activity, but instead reports on immediate changes in neuronal activity, which are evident after treatment with harmaline (*Herrera and Robertson, 1996*). The genetic manipulation to block Purkinje cell output results in altered firing properties throughout the life of the mice, and therefore the changes in cerebellar nuclei firing activity are constitutive. Altogether, these data suggest that Purkinje cell signaling is required for propagating the neural activity that drives harmaline induced tremor, and that Purkinje cell to cerebellar nuclei communication is an essential synapse for promoting tremor behavior in mice.

## Harmaline causes Purkinje cells to develop a bursting pattern of activity in awake behaving mice

As our data pointed to Purkinje cell activity as a primary factor in mediating tremor, we sought to define the underlying Purkinje cell activity that occurs during tremor. We performed single unit extracellular recordings of Purkinje cells in awake head-fixed mice both before and during tremor that was triggered by the acute effects of harmaline (*Figure 2a–c*). This allowed us to record and quantify both the Purkinje cells' simple spikes, which are both intrinsically generated and modulated by mossy fiber inputs, as well as complex spikes which are generated via the climbing fiber input (*Figures 1c* and *2c*). We recorded Purkinje cells in lobules IV, V, and VI of the vermis as well as Purkinje cells in the adjacent regions of the paravermis due to the involvement of these lobules in ongoing locomotion (*Valle et al., 2008*; *Valle et al., 2012*; *Armstrong and Edgley, 1988*; *Edgley and Lidierth, 1988*). Moreover, the anterior lobules process signals concerning the control of limb movements, which are severely affected in harmaline tremor (*Kuo et al., 2019*). In both control and mutant animals, Purkinje cell simple spike activity developed a dramatic bursting pattern during tremor (*Figure 2d–g*; also see *White et al., 2014* for a characterization of the mutant cerebellar activity). When we quantified the spike properties of these cells, we found that Purkinje cell simple spike activity had a significantly decreased frequency and significantly increased coefficient of variance (CV) and CV2 during tremor (*Holt et al., 1996*; *Figure 2h–j*; *Table 2*; *Control baseline N = 4, n = 18. Control + harmaline N = 6, n = 14. Mutant baseline N = 4, n = 15. Mutant + harmaline N = 5, n = 12*). CV is a measure of irregularity of interspike intervals over the entire recording of the cell, and therefore a higher CV value indicates a greater overall bursting quality of cell activity. Meanwhile, CV2 is a measure of irregularity of directly adjacent interspike intervals, and therefore a higher CV2 value indicates a more erratic and unpredictable quality of a spike train. However, CV2 can also be elevated when there is an overall bursting quality of cell activity, especially when the number of spikes within a burst or the overall firing rate is low, as we have shown here (*Figure 2e and g–h*; *Holt et al., 1996*). Therefore, simple spike activity predominantly decreases in frequency and increases in 'burstiness' after harmaline administration.

Complex spike properties changed in the opposite direction, wherein frequency was significantly increased and CV2 significantly decreased, while CV was not significantly altered in control animals (*Figure 2k–m*; *Table 2*; *Control baseline N = 4, n = 18. Control + harmaline N = 6, n = 14. Mutant baseline N = 4, n = 15. Mutant + harmaline N = 5, n = 12*). Bursts of activity tended to be led by a complex spike and preceded by a pause in activity (*Figure 2e and g*). This is evidenced by a significantly greater duration of inter-spike interval (ISI) before each complex spike and no change in duration after each complex spike after harmaline administration (*Figure 2n–o*; *Table 2*; *Control baseline N = 4, n = 18. Control + harmaline N = 6, n = 14. Mutant baseline N = 4, n = 15. Mutant + harmaline N = 5, n = 12*). Therefore, complex spike timing properties also apply to burst timing properties. Since simple spike frequency decreased and the complex spike frequency increased, this resulted in an overall decrease in simple spike to complex spike ratio during tremor (*Figure 2p*; *Table 2*;

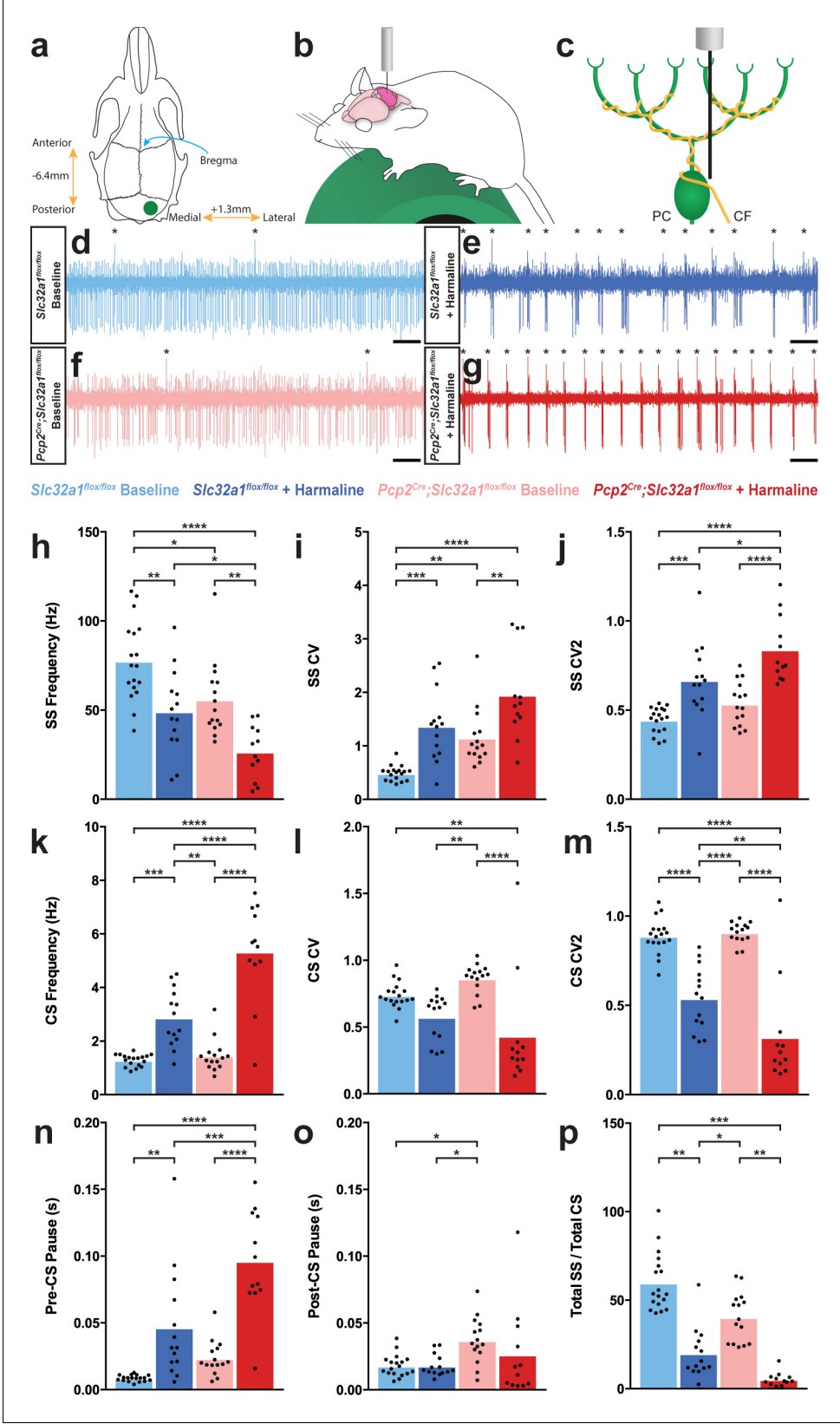

**Figure 2.** Purkinje cell firing patterns are significantly altered after harmaline administration. (a) Representation of craniotomy site using skull landmarks. Craniotomy (green circle) for awake head-fixed neural recordings was made −6.4 mm from Bregma (blue arrow) and 1.3 mm lateral from midline. (b) Representation of awake head-fixed

*Figure 2 continued on next page*

*Figure 2 continued*

recordings. Mice were allowed to stand on a green foam wheel (green cylinder) during recordings. (c) Representation of extracellular recordings of Purkinje cells (PC) which allowed recordings of simple spikes and complex spikes, which are triggered by the climbing fiber (CF). (d–g) Example raw traces from recordings of Purkinje cells. Complex spikes are indicated with asterisks. Scale = 500 ms. (d) Purkinje cell from a control animal. (e) Purkinje cell from a control animal during tremor after harmaline administration. (f) Purkinje cell from a mutant animal. (g) Purkinje cell from a mutant animal after harmaline administration (tremor not present). (h–j) Quantification of Purkinje cell simple spike firing properties including frequency (h), CV (i), and CV2 (j). (k–m) Quantification of Purkinje cell complex spike firing properties including frequency (k), CV (l), and CV2 (m). (n–p) Quantification of Purkinje cell simple spike and complex spike relationship including pre complex spike pause duration (n), post complex spike pause duration (o), and total simple spike to complex spike ratio (p). Source data for **h–p** are available in *Figure 2—source data 1*.

The online version of this article includes the following source data for figure 2:

**Source data 1.** Source data for representative graphs in *Figure 2*.

---

*Control baseline N = 4, n = 18. Control + harmaline N = 6, n = 17. Mutant baseline N = 4, n = 18. Mutant + harmaline N = 5, n = 15*). Together, these data indicate that Purkinje cell output is defined by an overall lower simple spike firing rate and a steady pattern of simple spike bursts after harmaline administration in an awake behaving condition. These bursts are flanked by complex spikes that are increased in frequency and regularity. In all quantified measures of spike activity, a similar directionality of change occurred in the mutant Purkinje cells of mice with silenced Purkinje cell output compared to those in the controls. These electrophysiological data suggest that if the circuits that carry tremor related neural activity eventually innervate the Purkinje cells, then the fidelity of the pathways that transfer the signals is equivalent in mutants and controls since both genotypes of mice had similar in vivo neuronal responses to the drug (*Figure 2h–p*).

## Purkinje cell neurotransmission is necessary for propagating harmaline-induced burst activity

We next tested what effect the abnormal Purkinje cell activity has on the downstream target neurons in the cerebellar nuclei because these cells provide the major output of the cerebellum. Of the three cerebellar nuclei, we focused our attention on the middle one called the interposed nucleus since it is critical for ongoing movement (*Becker and Person, 2019*; *Low et al., 2018*; *Bracha et al., 1999*), which is especially important in cerebellar-dependent tremors as the shaking behavior is usually evident during motion. We performed single unit extracellular recordings of cerebellar nuclei cells in awake head-fixed mice both before and during tremor caused by harmaline (*Figure 3a*). We found significantly different responses in the cerebellar nuclei of control mice compared to those lacking Purkinje cell signaling (*Figure 3b–e*). Cerebellar nuclei cells in control animals that were experiencing tremor had a significant and more predominant bursting pattern as measured by CV, with no significant change in firing frequency or CV2 compared to baseline firing (*Figure 3f–h*, *Table 3*). Importantly, the vast majority of cerebellar nuclei cells recorded during tremor in the treated control animals exhibited this bursting neuronal activity. Analysis of the recorded cells revealed that all but one of the cerebellar nuclei cells recorded from control animals during tremor were calculated to have a CV value greater than the mean of the control group at baseline. Additionally, 57.14% of cells had a greater CV during tremor than the maximum recorded CV of cellular activity at baseline. However, cerebellar nuclei cells in animals lacking Purkinje cell GABA neurotransmission – which results in little to no tremor after harmaline administration – had no change from baseline in any of our measures of cerebellar nuclei spike properties (*Figure 3f–h*; *Table 3*; *Control baseline N = 5, n = 19. Control + harmaline N = 3, n = 14. Mutant baseline N = 6, n = 18. Mutant + harmaline N = 4, n = 11*). These data suggest that in vivo circuit alterations that promote abnormal burst activity in the cerebellar nuclei, with Purkinje cell signals as one major source, could drive the core behavioral features of tremor in behaving mice.

**Table 2.** Precision measures, exact p-values, and replicate data relevant to *Figure 2*.

| Figure | Comparator 1 | Comparator 2 | Mean 1 | Mean 2 | SE of diff. | N 1 | N 2 | Summary | Adjusted P Value |
|---|---|---|---|---|---|---|---|---|---|
| *Figure 2h* | *Slc32a1*^flox/flox baseline | *Slc32a1*^flox/flox + harmaline | 77.63 Hz | 49.18 Hz | 7.497 | 18 | 14 | ** | 0.0020 |
| | *Slc32a1*^flox/flox baseline | *Pcp2*^Cre;*Slc32a1*^flox/flox baseline | 77.63 Hz | 55.97 Hz | 7.355 | 18 | 15 | * | 0.0238 |
| | *Slc32a1*^flox/flox baseline | *Pcp2*^Cre;*Slc32a1*^flox/flox + harmaline | 77.63 Hz | 26.65 Hz | 7.840 | 18 | 12 | **** | <0.0001 |
| | *Slc32a1*^flox/flox + harmaline | *Pcp2*^Cre;*Slc32a1*^flox/flox baseline | 49.18 Hz | 55.97 Hz | 7.818 | 14 | 15 | ns | 0.8208 |
| | *Slc32a1*^flox/flox + harmaline | *Pcp2*^Cre;*Slc32a1*^flox/flox + harmaline | 49.18 Hz | 26.65 Hz | 8.276 | 14 | 12 | * | 0.0419 |
| | *Pcp2*^Cre;*Slc32a1*^flox/flox baseline | *Pcp2*^Cre;*Slc32a1*^flox/flox + harmaline | 55.97 Hz | 26.65 Hz | 8.148 | 15 | 12 | ** | 0.0037 |
| *Figure 2i* | *Slc32a1*^flox/flox baseline | *Slc32a1*^flox/flox + harmaline | 0.4902 | 1.372 | 0.2017 | 18 | 14 | *** | 0.0003 |
| | *Slc32a1*^flox/flox baseline | *Pcp2*^Cre;*Slc32a1*^flox/flox baseline | 0.4902 | 1.152 | 0.1979 | 18 | 15 | ** | 0.0079 |
| | *Slc32a1*^flox/flox baseline | *Pcp2*^Cre;*Slc32a1*^flox/flox + harmaline | 0.4902 | 1.953 | 0.2109 | 18 | 12 | **** | <0.0001 |
| | *Slc32a1*^flox/flox + harmaline | *Pcp2*^Cre;*Slc32a1*^flox/flox baseline | 1.372 | 1.152 | 0.2103 | 14 | 15 | ns | 0.7229 |
| | *Slc32a1*^flox/flox + harmaline | *Pcp2*^Cre;*Slc32a1*^flox/flox + harmaline | 1.372 | 1.953 | 0.2227 | 14 | 12 | ns | 0.0552 |
| | *Pcp2*^Cre;*Slc32a1*^flox/flox baseline | *Pcp2*^Cre;*Slc32a1*^flox/flox + harmaline | 1.152 | 1.953 | 0.2192 | 15 | 12 | ** | 0.0032 |
| *Figure 2j* | *Slc32a1*^flox/flox baseline | *Slc32a1*^flox/flox + harmaline | 0.4443 | 0.6683 | 0.05314 | 18 | 14 | *** | 0.0005 |
| | *Slc32a1*^flox/flox baseline | *Pcp2*^Cre;*Slc32a1*^flox/flox baseline | 0.4443 | 0.5355 | 0.05214 | 18 | 15 | ns | 0.3088 |
| | *Slc32a1*^flox/flox baseline | *Pcp2*^Cre;*Slc32a1*^flox/flox + harmaline | 0.4443 | 0.8396 | 0.05558 | 18 | 12 | **** | <0.0001 |
| | *Slc32a1*^flox/flox + harmaline | *Pcp2*^Cre;*Slc32a1*^flox/flox baseline | 0.6683 | 0.5355 | 0.05542 | 14 | 15 | ns | 0.0897 |
| | *Slc32a1*^flox/flox + harmaline | *Pcp2*^Cre;*Slc32a1*^flox/flox + harmaline | 0.6683 | 0.8396 | 0.05867 | 14 | 12 | * | 0.0253 |
| | *Pcp2*^Cre;*Slc32a1*^flox/flox baseline | *Pcp2*^Cre;*Slc32a1*^flox/flox + harmaline | 0.5355 | 0.8396 | 0.05776 | 15 | 12 | **** | <0.0001 |
| *Figure 2k* | *Slc32a1*^flox/flox baseline | *Slc32a1*^flox/flox + harmaline | 1.289 Hz | 2.876 Hz | 0.3641 | 18 | 14 | *** | 0.0003 |
| | *Slc32a1*^flox/flox baseline | *Pcp2*^Cre;*Slc32a1*^flox/flox baseline | 1.289 Hz | 1.458 Hz | 0.3572 | 18 | 15 | ns | 0.9649 |
| | *Slc32a1*^flox/flox baseline | *Pcp2*^Cre;*Slc32a1*^flox/flox + harmaline | 1.289 Hz | 5.338 Hz | 0.3807 | 18 | 12 | **** | <0.0001 |
| | *Slc32a1*^flox/flox + harmaline | *Pcp2*^Cre;*Slc32a1*^flox/flox baseline | 2.876 Hz | 1.458 Hz | 0.3797 | 14 | 15 | ** | 0.0025 |
| | *Slc32a1*^flox/flox + harmaline | *Pcp2*^Cre;*Slc32a1*^flox/flox + harmaline | 2.876 Hz | 5.338 Hz | 0.4019 | 14 | 12 | **** | <0.0001 |
| | *Pcp2*^Cre;*Slc32a1*^flox/flox baseline | *Pcp2*^Cre;*Slc32a1*^flox/flox + harmaline | 1.458 Hz | 5.338 Hz | 0.3957 | 15 | 12 | **** | <0.0001 |

*Table 2 continued on next page*

*Table 2 continued*

| Figure | Comparator 1 | Comparator 2 | Mean 1 | Mean 2 | SE of diff. | N 1 | N 2 | Summary | Adjusted P Value |
|---|---|---|---|---|---|---|---|---|---|
| *Figure 2l* | $Slc32a1^{flox/flox}$ baseline | $Slc32a1^{flox/flox}$ + harmaline | 0.7396 | 0.5753 | 0.07763 | 18 | 14 | ns | 0.1609 |
| | $Slc32a1^{flox/flox}$ baseline | $Pcp2^{Cre};Slc32a1^{flox/flox}$ baseline | 0.7396 | 0.8631 | 0.07617 | 18 | 15 | ns | 0.3751 |
| | $Slc32a1^{flox/flox}$ baseline | $Pcp2^{Cre};Slc32a1^{flox/flox}$ + harmaline | 0.7396 | 0.4335 | 0.08119 | 18 | 12 | ** | 0.0022 |
| | $Slc32a1^{flox/flox}$ + harmaline | $Pcp2^{Cre};Slc32a1^{flox/flox}$ baseline | 0.5753 | 0.8631 | 0.08096 | 14 | 15 | ** | 0.0043 |
| | $Slc32a1^{flox/flox}$ + harmaline | $Pcp2^{Cre};Slc32a1^{flox/flox}$ + harmaline | 0.5753 | 0.4335 | 0.08571 | 14 | 12 | ns | 0.3572 |
| | $Pcp2^{Cre};Slc32a1^{flox/flox}$ baseline | $Pcp2^{Cre};Slc32a1^{flox/flox}$ + harmaline | 0.8631 | 0.4335 | 0.08438 | 15 | 12 | **** | <0.0001 |
| *Figure 2m* | $Slc32a1^{flox/flox}$ baseline | $Slc32a1^{flox/flox}$ + harmaline | 0.8876 | 0.5399 | 0.05937 | 18 | 14 | **** | <0.0001 |
| | $Slc32a1^{flox/flox}$ baseline | $Pcp2^{Cre};Slc32a1^{flox/flox}$ baseline | 0.8876 | 0.9082 | 0.05824 | 18 | 15 | ns | 0.9848 |
| | $Slc32a1^{flox/flox}$ baseline | $Pcp2^{Cre};Slc32a1^{flox/flox}$ + harmaline | 0.8876 | 0.3224 | 0.06209 | 18 | 12 | **** | <0.0001 |
| | $Slc32a1^{flox/flox}$ + harmaline | $Pcp2^{Cre};Slc32a1^{flox/flox}$ baseline | 0.5399 | 0.9082 | 0.06191 | 14 | 15 | **** | <0.0001 |
| | $Slc32a1^{flox/flox}$ + harmaline | $Pcp2^{Cre};Slc32a1^{flox/flox}$ + harmaline | 0.5399 | 0.3224 | 0.06554 | 14 | 12 | ** | 0.0085 |
| | $Pcp2^{Cre};Slc32a1^{flox/flox}$ baseline | $Pcp2^{Cre};Slc32a1^{flox/flox}$ + harmaline | 0.9082 | 0.3224 | 0.06453 | 15 | 12 | **** | <0.0001 |
| *Figure 2n* | $Slc32a1^{flox/flox}$ baseline | $Slc32a1^{flox/flox}$ + harmaline | 0.008528 s | 0.04651 s | 0.009766 | 18 | 14 | ** | 0.0015 |
| | $Slc32a1^{flox/flox}$ baseline | $Pcp2^{Cre};Slc32a1^{flox/flox}$ baseline | 0.008528 s | 0.02349 s | 0.009581 | 18 | 15 | ns | 0.4089 |
| | $Slc32a1^{flox/flox}$ baseline | $Pcp2^{Cre};Slc32a1^{flox/flox}$ + harmaline | 0.008528 s | 0.09622 s | 0.01021 | 18 | 12 | **** | <0.0001 |
| | $Slc32a1^{flox/flox}$ + harmaline | $Pcp2^{Cre};Slc32a1^{flox/flox}$ baseline | 0.04651 s | 0.02349 s | 0.01018 | 14 | 15 | ns | 0.1202 |
| | $Slc32a1^{flox/flox}$ + harmaline | $Pcp2^{Cre};Slc32a1^{flox/flox}$ + harmaline | 0.04651 s | 0.09622 s | 0.01078 | 14 | 12 | *** | 0.0001 |
| | $Pcp2^{Cre};Slc32a1^{flox/flox}$ baseline | $Pcp2^{Cre};Slc32a1^{flox/flox}$ + harmaline | 0.02349 s | 0.09622 s | 0.01061 | 15 | 12 | **** | <0.0001 |
| *Figure 2o* | $Slc32a1^{flox/flox}$ baseline | $Slc32a1^{flox/flox}$ + harmaline | 0.01806 s | 0.01806 s | 0.006551 | 18 | 14 | ns | >0.9999 |
| | $Slc32a1^{flox/flox}$ baseline | $Pcp2^{Cre};Slc32a1^{flox/flox}$ baseline | 0.01806 s | 0.03709 s | 0.006427 | 18 | 15 | * | 0.0228 |
| | $Slc32a1^{flox/flox}$ baseline | $Pcp2^{Cre};Slc32a1^{flox/flox}$ + harmaline | 0.01806 s | 0.02643 s | 0.006851 | 18 | 12 | ns | 0.6161 |
| | $Slc32a1^{flox/flox}$ + harmaline | $Pcp2^{Cre};Slc32a1^{flox/flox}$ baseline | 0.01806 s | 0.03709 s | 0.006831 | 14 | 15 | * | 0.0358 |
| | $Slc32a1^{flox/flox}$ + harmaline | $Pcp2^{Cre};Slc32a1^{flox/flox}$ + harmaline | 0.01806 s | 0.02643 s | 0.007232 | 14 | 12 | ns | 0.6560 |
| | $Pcp2^{Cre};Slc32a1^{flox/flox}$ baseline | $Pcp2^{Cre};Slc32a1^{flox/flox}$ + harmaline | 0.03709 s | 0.02643 s | 0.007120 | 15 | 12 | ns | 0.4462 |

*Table 2 continued on next page*

*Table 2 continued*

| Figure | Comparator 1 | Comparator 2 | Mean 1 | Mean 2 | SE of diff. | N 1 | N 2 | Summary | Adjusted P Value |
|---|---|---|---|---|---|---|---|---|---|
| *Figure 2p* | $Slc32a1^{flox/flox}$ baseline | $Slc32a1^{flox/flox}$ + harmaline | 59.78 | 17.71 | 11.71 | 18 | 17 | ** | 0.0035 |
| | $Slc32a1^{flox/flox}$ baseline | $Pcp2^{Cre};Slc32a1^{flox/flox}$ baseline | 59.78 | 53.83 | 11.54 | 18 | 18 | ns | 0.9551 |
| | $Slc32a1^{flox/flox}$ baseline | $Pcp2^{Cre};Slc32a1^{flox/flox}$ + harmaline | 59.78 | 8.421 | 12.10 | 18 | 15 | *** | 0.0004 |
| | $Slc32a1^{flox/flox}$ + harmaline | $Pcp2^{Cre};Slc32a1^{flox/flox}$ baseline | 17.71 | 53.83 | 11.71 | 17 | 18 | * | 0.0155 |
| | $Slc32a1^{flox/flox}$ + harmaline | $Pcp2^{Cre};Slc32a1^{flox/flox}$ + harmaline | 17.71 | 8.421 | 12.26 | 17 | 15 | ns | 0.8730 |
| | $Pcp2^{Cre};Slc32a1^{flox/flox}$ baseline | $Pcp2^{Cre};Slc32a1^{flox/flox}$ + harmaline | 53.83 | 8.421 | 12.10 | 18 | 15 | ** | 0.0021 |

## Rhythmic bursting activity in the cerebellar nuclei produces tremor across a range of frequencies

The genetic manipulation of Purkinje cells in the absence and presence of harmaline treatment argues that, whereas losing Purkinje cell activity is not a driver for tremor, changing their pattern and mode of interaction with the cerebellar nuclei might be. We subsequently tested whether recreating the abnormal cerebellar activity found in control mice treated with harmaline is sufficient for producing tremor. We implanted optical fibers bilaterally into the interposed nuclei of

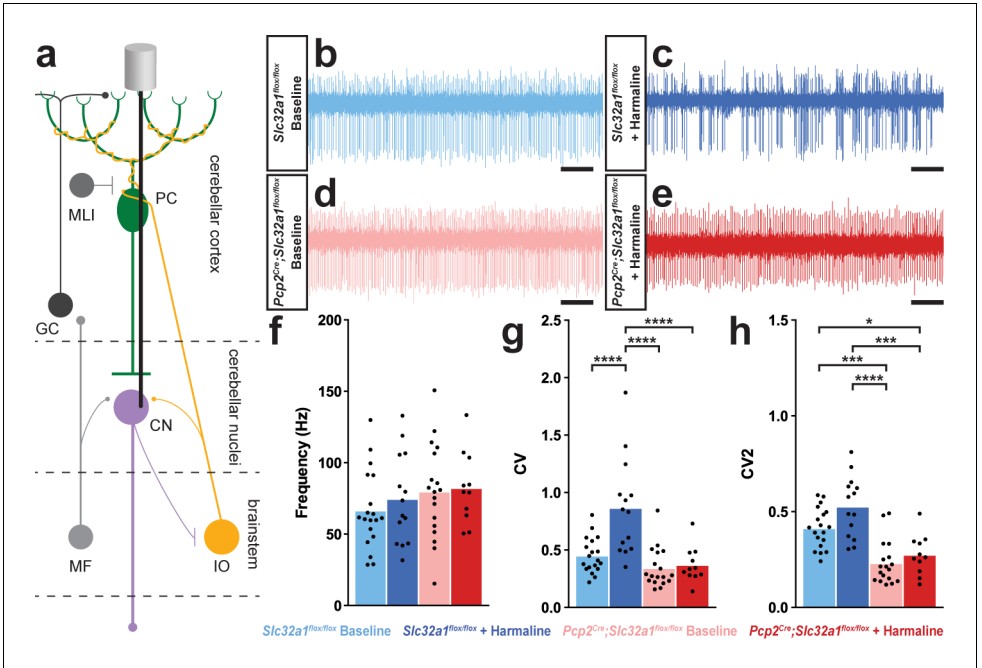

**Figure 3.** A burst pattern of cerebellar nuclei activity is associated with the tremor phenotype. (a) Representation of an extracellular recording of the cerebellar nuclei. (b–e) Example raw traces from recordings of cerebellar nuclei cells. Scale = 250 ms. (b) Cerebellar nuclei cell from a control animal. (c) Cerebellar nuclei cell from a control animal during tremor, after harmaline administration. (d) Cerebellar nuclei cell from a mutant animal. (e) Cerebellar nuclei cell from a mutant animal after harmaline administration (tremor not present). (f) Quantification of population cerebellar nuclei firing frequency. (g) Quantification of population cerebellar nuclei CV. (h) Quantification of population cerebellar nuclei CV2. Source data for f–h are available in *Figure 3—source data 1*. The online version of this article includes the following source data for figure 3:

**Source data 1.** Source data for representative graphs in *Figure 3*.

**Table 3.** Precision measures, exact p-values, and replicate data relevant to *Figure 3*.

| Figure | Comparator 1 | Comparator 2 | Mean 1 | Mean 2 | SE of diff. | N 1 | N 2 | Summary | Adjusted P Value |
|---|---|---|---|---|---|---|---|---|---|
| *Figure 3f* | *Slc32a1^flox/flox* baseline | *Slc32a1^flox/flox* + harmaline | 66.48 Hz | 74.56 Hz | 10.44 | 19 | 14 | ns | 0.8660 |
| | *Slc32a1^flox/flox* baseline | *Pcp2^Cre;Slc32a1^flox/flox* baseline | 66.48 Hz | 79.81 Hz | 9.753 | 19 | 18 | ns | 0.5246 |
| | *Slc32a1^flox/flox* baseline | *Pcp2^Cre;Slc32a1^flox/flox* + harmaline | 66.48 Hz | 82.22 Hz | 11.23 | 19 | 11 | ns | 0.5035 |
| | *Slc32a1^flox/flox* + harmaline | *Pcp2^Cre;Slc32a1^flox/flox* baseline | 74.56 Hz | 79.81 Hz | 10.57 | 14 | 18 | ns | 0.9593 |
| | *Slc32a1^flox/flox* + harmaline | *Pcp2^Cre;Slc32a1^flox/flox* + harmaline | 74.56 Hz | 82.22 Hz | 11.95 | 14 | 11 | ns | 0.9180 |
| | *Pcp2^Cre;Slc32a1^flox/flox* baseline | *Pcp2^Cre;Slc32a1^flox/flox* + harmaline | 79.81 Hz | 82.22 Hz | 11.35 | 18 | 11 | ns | 0.9966 |
| *Figure 3g* | *Slc32a1^flox/flox* baseline | *Slc32a1^flox/flox* + harmaline | 0.4511 | 0.8654 | 0.08621 | 19 | 14 | **** | <0.0001 |
| | *Slc32a1^flox/flox* baseline | *Pcp2^Cre;Slc32a1^flox/flox* baseline | 0.4511 | 0.3428 | 0.08050 | 19 | 18 | ns | 0.5379 |
| | *Slc32a1^flox/flox* baseline | *Pcp2^Cre;Slc32a1^flox/flox* + harmaline | 0.4511 | 0.3695 | 0.09273 | 19 | 11 | ns | 0.8150 |
| | *Slc32a1^flox/flox* + harmaline | *Pcp2^Cre;Slc32a1^flox/flox* baseline | 0.8654 | 0.3428 | 0.08722 | 14 | 18 | **** | <0.0001 |
| | *Slc32a1^flox/flox* + harmaline | *Pcp2^Cre;Slc32a1^flox/flox* + harmaline | 0.8654 | 0.3695 | 0.09861 | 14 | 11 | **** | <0.0001 |
| | *Pcp2^Cre;Slc32a1^flox/flox* baseline | *Pcp2^Cre;Slc32a1^flox/flox* + harmaline | 0.3428 | 0.3695 | 0.09367 | 18 | 11 | ns | 0.9918 |
| *Figure 3h* | *Slc32a1^flox/flox* baseline | *Slc32a1^flox/flox* + harmaline | 0.4141 | 0.5262 | 0.04266 | 19 | 14 | ns | 0.0520 |
| | *Slc32a1^flox/flox* baseline | *Pcp2^Cre;Slc32a1^flox/flox* baseline | 0.4141 | 0.2310 | 0.03984 | 19 | 18 | *** | 0.0001 |
| | *Slc32a1^flox/flox* baseline | *Pcp2^Cre;Slc32a1^flox/flox* + harmaline | 0.4141 | 0.2743 | 0.04589 | 19 | 11 | * | 0.0179 |
| | *Slc32a1^flox/flox* + harmaline | *Pcp2^Cre;Slc32a1^flox/flox* baseline | 0.5262 | 0.2310 | 0.04316 | 14 | 18 | **** | <0.0001 |
| | *Slc32a1^flox/flox* + harmaline | *Pcp2^Cre;Slc32a1^flox/flox* + harmaline | 0.5262 | 0.2743 | 0.04880 | 14 | 11 | **** | <0.0001 |
| | *Pcp2^Cre;Slc32a1^flox/flox* baseline | *Pcp2^Cre;Slc32a1^flox/flox* + harmaline | 0.2310 | 0.2743 | 0.04636 | 18 | 11 | ns | 0.7864 |

*Pcp2^Cre;ROSA26^loxP-STOP-loxP-EYFP-ChR2* mice in which the opsin is only expressed in Purkinje cells (*Figure 4—figure supplement 1*). Similar to the electrophysiology recordings in the harmaline treated mice, we again focused our attention on the interposed nuclei because of their role during ongoing movement. We also implanted EMG electrodes into the gastrocnemius of the left hind limb to measure tremor with a particular emphasis on examining muscle activity that occurs during movements (*Figure 4a–c*). We initially used anesthetized mice to test whether activating ChR2-expressing Purkinje cell terminals induces an inhibition of interposed nuclear neurons by simultaneously recording interposed neurons and optogenetically stimulating Purkinje cell terminals with glass electrodes outfitted with an internal optic fiber (*Figure 4d–j*). Next, we stimulated Purkinje cell terminals with sinusoidal pulses of light at 1–20 Hz to induce different frequencies of bursting activity of the cerebellar nuclei. The stimulation in the cerebellum was bilateral in order to induce a balanced disturbance in gait on both sides of the body, although EMG was only conducted on one side to minimize discomfort and avoid confounding the muscle activity measurements taken during quiet wakefulness and motion. We found that inducing bursting patterns of cerebellar nuclei activity resulted in tremor (*Figure 4k–l*; *Video 3*). Tremor could be elicited at all frequencies tested and was visible by eye and detectable in the EMG trace (*Figure 4k–p*). The predominant frequency of tremor elicited matched the frequency of optogenetic stimulation and was only present during stimulation (*Figure 4m–o*; *Table 4*; *Figure 4n*; N = 7). Interestingly, the optogenetically-elicited tremor was not uniform in

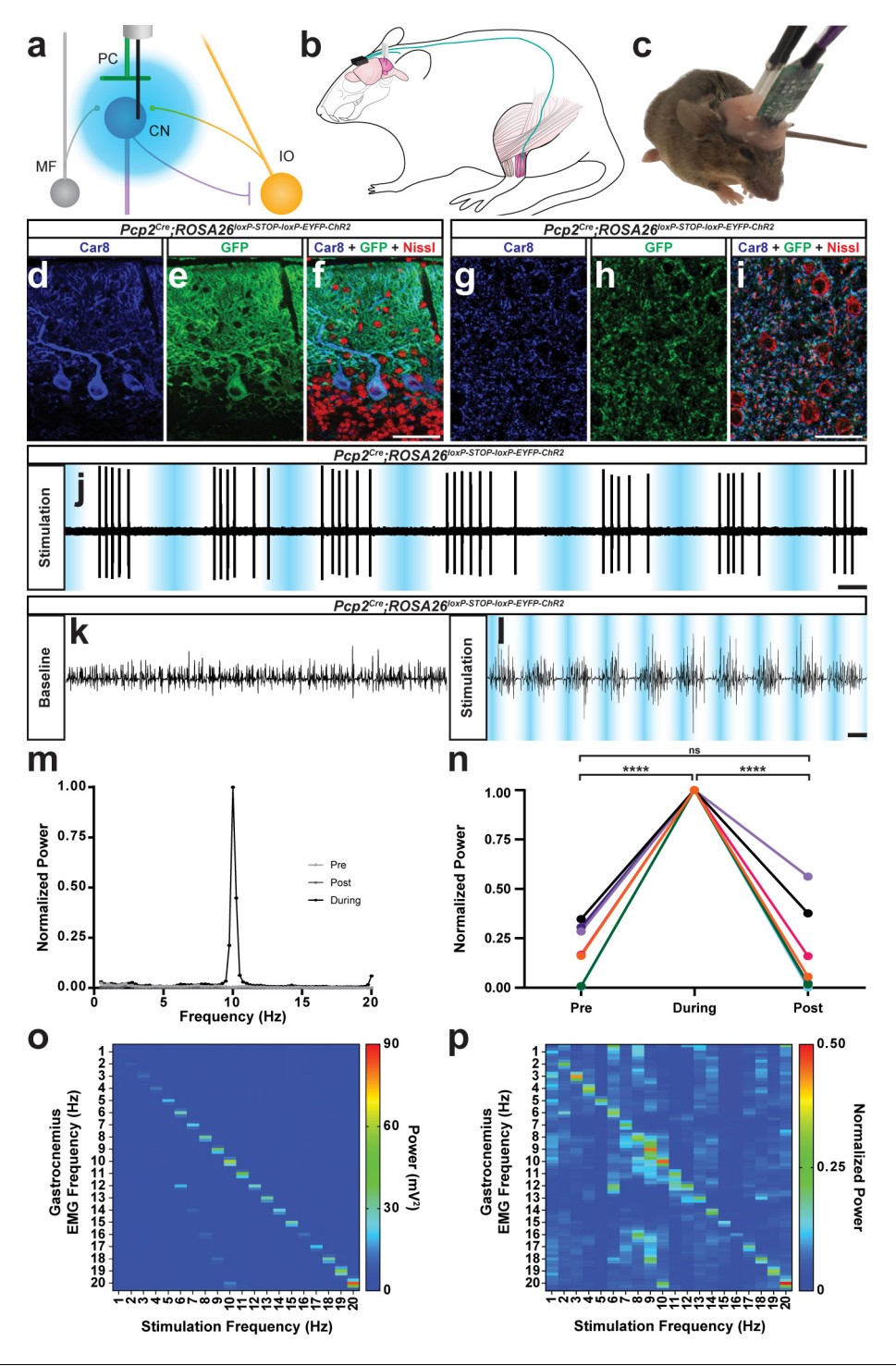

**Figure 4.** The burst pattern of cerebellar nuclei activity is sufficient to produce tremor at multiple frequencies. (a) Representation of an optical fiber in the cerebellar nuclei. (b) Representation of EMG and optical fiber implant strategy. EMG electrodes are implanted into the gastrocnemius muscle (pink) of the left hindlimb and electrode wire (teal) is fed under the skin to a connector (black) fixed to the skull. An additional wire is fed from the connector to under the skin of the neck region as a ground. Two optical fibers (white) are implanted bilaterally targeting the interposed nucleus. (c) Image of a mouse during an EMG recording. A preamplifier (green) is placed in the connector and tethered to a passive commutator (not pictured). Fiber patch cables (black cables) are connected to the implanted optical fibers. (d–f) Triple-stained fluorescent micrographs of the cerebellar cortex.

*Figure 4 continued on next page*

*Figure 4 continued*

Scale = 50 µm. (**d**) CAR8 protein in Purkinje cells stained blue. (**e**) Membrane-bound ChR2 labeled with green fluorescent protein (GFP) stained green. (**f**) Composite of CAR8 (blue), GFP (green), and Nissl to label all cells (red). (**g–i**) Triple-stained fluorescent micrograph of cerebellar nuclei cells. Scale = 50 µm. (**g**) CAR8 protein in Purkinje cell terminals in the cerebellar nuclei (blue). (**h**) Membrane-bound ChR2 labeled with GFP (green). (**d**) Composite of CAR8 (blue), GFP (green), and Nissl to label all cells (red). (**j**) Example extracellular recording from a cerebellar nuclei cell during ChR2 stimulation of surrounding Purkinje cell terminals (blue bars). Scale = 250 ms. (**k–l**) Example raw EMG traces from a *Pcp2^Cre;ROSA26^loxP-STOP-loxP-ChR2-EYFP* mouse. Scale = 50 ms. (**k**) Baseline EMG trace before optogenetic stimulation. (**l**) EMG trace during tremor caused by optogenetic stimulation. Stimulation periods indicated by overlaid blue bars. (**m**) Example power spectrum analysis from an animal receiving optogenetic stimulation at 10 Hz, normalized to peak tremor power. Pre = pre stimulation period (baseline). Post = post stimulation period. During = during stimulation period. (**n**) EMG power at 10 Hz during 10 Hz stimulation for all mice tested, normalized to each individual's overall maximum power in the pre, during, and post stimulation periods. Pre vs during stimulation period: p<0.0001. During vs post stimulation period: p=0.0001. Pre vs post stimulation period: p=0.9796, not significant (ns). N = 7. (**o**) Heat plot showing population average elicited EMG power for each optogenetic stimulation frequency tested. Heat scale = 0 to 90 mV$^2$. (**p**) Heat plot showing population average of power normalized to individual peaks for each optogenetic stimulation frequency tested. Heat scale = 0 to 0.5. Source data for **m–p** are available in *Figure 4—source data 1*.

The online version of this article includes the following source data and figure supplement(s) for figure 4:

**Source data 1.** Source data for representative graphs in *Figure 4*.

**Figure supplement 1.** Expression of channelrhodopsin is limited to Purkinje cells when reporter expression is driven with a *Pcp2^Cre* allele.

**Figure supplement 2.** Targeting of optic fibers for optogenetic stimulation of Purkinje cell terminals in the cerebellar nuclei.

**Figure supplement 3.** Tremor severity increases as optogenetic stimulation light power increases.

**Figure supplement 3—source data 1.** Source data for representative graphs in *Figure 4—figure supplement 3*.

---

severity across frequencies, with the maximum power occurring during tremor frequencies around 10 Hz and 20 Hz (*Figure 4o–p*). As all stimulation frequencies resulted in a strong and visible tremor, similarly when quantified nearly all stimulation-period tremor powers were found to be significantly greater than baseline tremor at the same frequency (1 Hz: p<0.0001, 2 Hz: p=0.0051, 3 Hz: p<0.0001, 4 Hz: p=0.0039, 5 Hz: p=0.0083, 6 Hz: p<0.0001, 7 Hz: p=0.0003, 8 Hz: p<0.0001, 9 Hz: p<0.0001, 10 Hz: p=0.001, 11 Hz: p=0.0083, 12 Hz: p=0.0219, 13 Hz: p=0.0304, 14 Hz: p=0.0029, 15 Hz: p=0.8285, 16 Hz: p=0.0304, 17 Hz: p<0.0001, 18 Hz: p=0.0004, 19 Hz: p<0.0001, 20 Hz: p=0.0219). These data show that Purkinje-cell-induced bursting of the cerebellar nuclei is capable of generating a phenotypically obvious tremor at a wide range of frequencies. However, the data also indicate that there may be an ideal band of frequencies and harmonics at which the cerebellum instigates the strongest responses in the muscle, which ultimately execute the tremor behavior.

We further tested the extent of the optogenetic stimulation necessary to induce tremor. Using a custom-built tremor monitor to measure the responses (*Figure 5a*), we varied the intensity of the light delivered to the interposed nucleus (*Figure 4—figure supplement 2*) at a single stimulation frequency and then systematically recorded the resulting tremor. Tremor severity increased exponentially with the intensity of the light, ranging from a mild intermittent tremor to a constant violent tremor (N = 4; linear regression of the natural log: R square = 0.6861, p<0.0001 *Figure 4—figure supplement 3*). We then performed freehand dissection and removal of the neural tissue to expose the optical fiber implant while it was still embedded in the tissue. This allowed us to examine the accuracy of targeting the nuclei and also provided an opportunity to test the spread of light from the intact

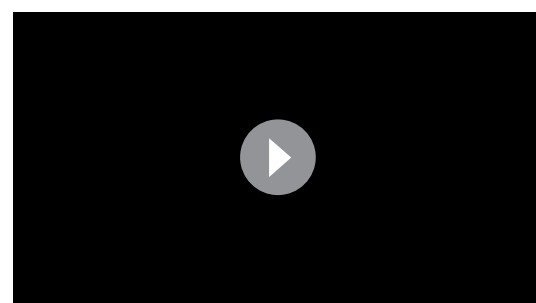

**Video 3.** Optogenetic stimulation of Purkinje cell terminals in the interposed nuclei of *Pcp2^Cre; ROSA26^loxP-STOP-loxP-ChR2-EYFP* mice using a 10 Hz sinusoidal stimulus induces a robust tremor.
https://elifesciences.org/articles/51928#video3

**Table 4.** Precision measures, exact p-values, and replicate data relevant to *Figure 4*.

| Figure | Comparator 1 | Comparator 2 | Mean 1 | Mean 2 | SE of diff. | N 1 | N 2 | Summary | Adjusted P Value |
|--------|--------------|--------------|--------|--------|-------------|-----|-----|---------|------------------|
| *Figure 4n* | pre | during | 0.1825 | 1.000 | 0.05238 | 7 | 7 | **** | <0.0001 |
| | pre | post | 0.1825 | 0.1702 | 0.06337 | 7 | 7 | ns | 0.9796 |
| | during | post | 1.000 | 0.1702 | 0.08224 | 7 | 7 | *** | 0.0001 |

optical fiber implant into the neural tissue. We found that at maximum light intensity the strongest optogenetic stimulation was targeted to the interposed nucleus (*Figure 4—figure supplement 2*). We further calculated using the light power capable of driving tremor that likely effective neurostimulation reached neurons at a depth of about 0.3 to 0.8 mm away from the fiber tip, using the threshold of 1 mW/mm$^2$ (*Deisseroth, 2012*; *Yizhar et al., 2011*).

## DBS directed to the cerebellar nuclei reduces tremor severity

Finally, because the data suggested that erroneous Purkinje cell to cerebellar nuclei communication was critical for the production of tremor, we aimed to correct this communication using deep brain stimulation (DBS) in order to treat ongoing tremor. We designed and built an open tremor monitor setup that would allow DBS cables to be connected to external equipment (*Figure 5a*; *Park et al., 2010*). We then devised a closed-loop DBS protocol that would constantly monitor the tremor behavior of the mouse and only trigger therapeutic stimulation during periods wherein the power of tremor was above a set threshold based on the animal's baseline physiological tremor recording (*Figure 5b*). We chose to use a high-frequency DBS approach (>100 Hz) based on the success of this therapeutic frequency range in human tremor diseases (*Miterko et al., 2018*; *Wilkes et al., 2020*). DBS was directed bilaterally to the interposed (middle) nuclei because they provide substantial output directly to the thalamus (*Low et al., 2018*; *Haroian et al., 1981*; *Gornati et al., 2018*; *Aumann et al., 1994*; *Stanton, 1980*), a brain region linked to tremor, and since this cerebellar nucleus is critical for ongoing motion, a physiological state that is particularly sensitive to tremors that involve the cerebellum. Moreover, the VIM and VAL regions of the thalamus, which interact with the cerebellum, are effective targets for DBS and lesion-based therapies in human essential tremor (*Wilkes et al., 2020*; *Baizabal-Carvallo et al., 2014*; *Zhang et al., 2010*; *Deuschl and Bain, 2002*). To start, we first allowed the animals to acclimate to the tremor monitor setup and then made baseline tremor recordings (*Figure 5c*). We then administered harmaline to induce tremor and allowed 15 min for the tremor to develop before moving to the DBS phase of the experiment (*Figure 5d*). Robust tremor was elicited by harmaline in implanted control animals, similar to our previous results (*Figures 5e* and *1f*, *Figure 5—figure supplement 1*). Closed-loop DBS was able to reduce tremor severity every time the threshold was crossed, was automatically shut off at levels below threshold, and then was triggered again on simultaneous bouts throughout the paradigm (*Figure 5f–g*, *Video 4*). We quantified tremor in short 80 s windows of time towards the end of the baseline period, after tremor had developed, and directly after activating stimulation. We normalized to the maximum power over the three time periods and found that tremor severity peaked during the harmaline period before stimulation was initiated and that tremor severity was decreased to levels that were not significantly different from baseline when the closed-loop DBS was activated (*Figure 5h–i*; *Table 5*; *N = 8*). These data show that therapeutic neuromodulation of cerebellar nuclei activity reduces tremor severity in behaving mice.

## Discussion

The participation of cerebellar dysfunction in a wide range of tremor disorders is universally anticipated (*Bhatia et al., 2018*). Here, we show that cerebellar neurons can produce the neural signals and behavioral states that are indicative of tremor. We demonstrate that it is not a lack of Purkinje cell activity, but instead an abnormal pattern of cerebellar output firing that causes tremor. Further, closed-loop DBS targeted to the cerebellar nuclei is sufficient to reduce pathological tremor.

While a loss of Purkinje cells and their ability to communicate with their downstream partners has been found in tremor disorders (*Louis, 2016*), we suggest here that pathological Purkinje cell activity

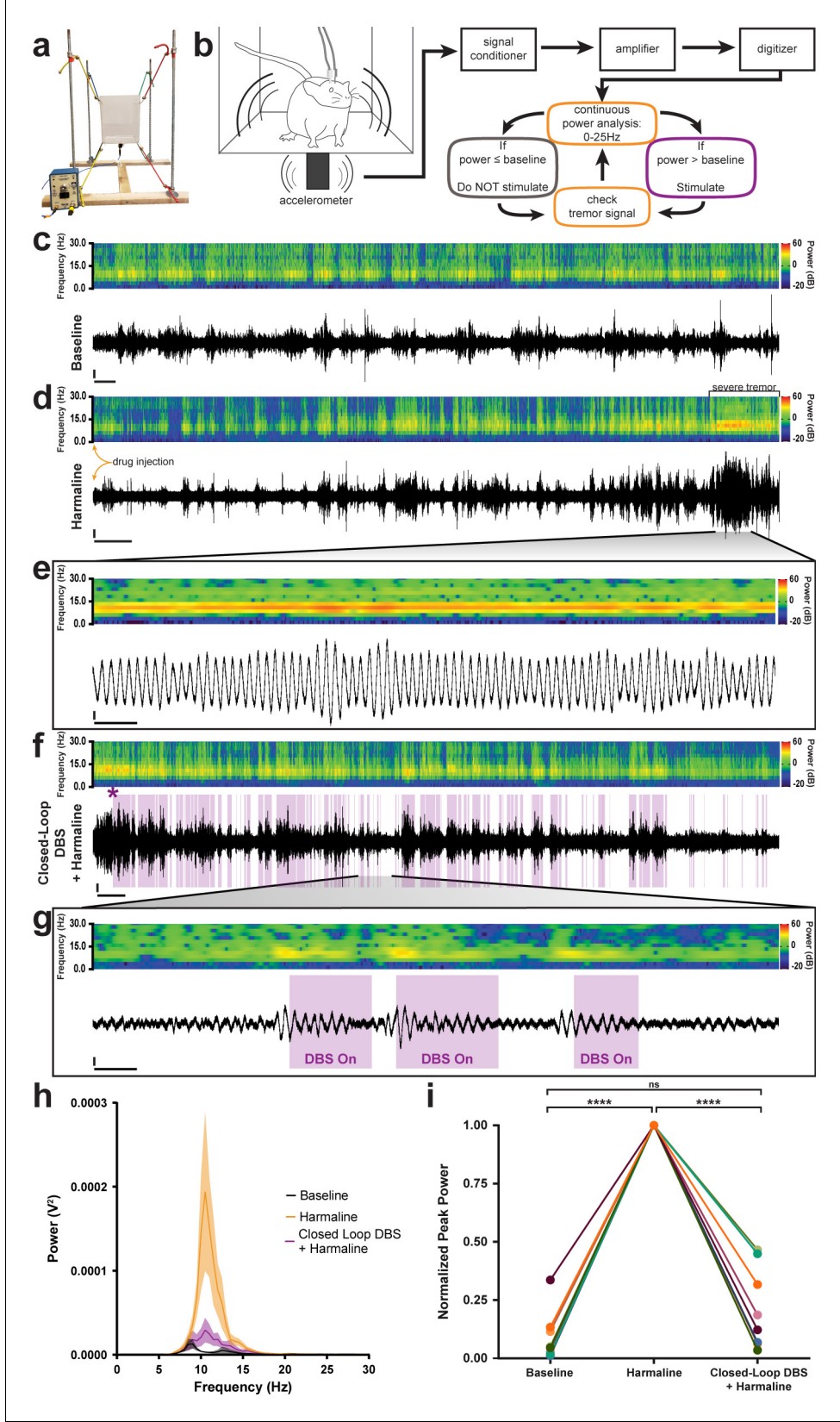

**Figure 5.** Closed-loop DBS targeted to the interposed nucleus reduces tremor severity. (**a**) Picture of the custom-built tremor monitor. (**b**) Representation of the closed-loop design. Left: a mouse exhibiting tremor behavior

*Figure 5 continued on next page*

*Figure 5 continued*

shakes the tremor monitor chamber with an attached accelerometer (black rectangle). Accelerometer signals are passed through a signal conditioner, amplifier, and digitizer before a continuous power spectrum analysis is applied for frequencies from 0 Hz to 25 Hz. If tremor power exceeds a set threshold based on the individual's baseline recording, stimulation is initiated for 36 ms before re-evaluation of tremor power. No stimulation is generated if tremor power is below threshold level. (c–g) Example tremor traces from a single animal. Top: sonogram view (continuous power spectrum) of raw tremor trace. All heat scales = −20 dB to 60 dB. Bottom: raw tremor traces. All vertical scales = 25 mV. (c) Baseline physiological tremor recording. Horizontal scale = 10 s. (d) Tremor recording beginning directly after harmaline injection (yellow arrows). Severe tremor develops as shown at the right of the recording (black bracket). Horizontal scale = 50 s. (e) Inset region from d (black lines) highlighting continuous, severe tremor. Horizontal scale = 0.5 s. (f) Tremor recording beginning directly after d and just before closed-loop DBS protocol is initiated. Initiation time indicated by purple asterisk. Stimulation periods indicated by overlaid purple bars. Horizontal scale = 50 s. (g) Inset region from f (black lines) highlighting bouts of tremor sufficient to cross threshold for stimulation. Stimulation periods indicated by overlaid purple bars. Horizontal scale = 0.5 s. (h) Population average tremor traces for baseline, harmaline tremor, and closed-loop DBS during harmaline tremor periods. Solid line = mean. Shaded region = SEM. N = 8. (i) Peak tremor power for each analyzed time period normalized to overall peak for all individuals. Power of tremor during harmaline period without closed-loop DBS stimulation was significantly greater than both baseline and the closed-loop DBS periods. There was no significant difference between baseline and the closed-loop DBS period. N = 8. Harmaline alone vs baseline: p<0.0001. Harmaline alone vs closed-loop DBS + harmaline: p<0.0001. Baseline vs closed-loop DBS + harmaline: p=0.3375. Source data for **h–i** are available in *Figure 5—source data 1*.

The online version of this article includes the following source data and figure supplement(s) for figure 5:

**Source data 1.** Source data for representative graphs in *Figure 5*.
**Figure supplement 1.** DBS electrodes targeted to the cerebellar interposed nucleus.

– whether before, in the context of, or without cell death – is a strong impetus of tremor. We first showed evidence for this using *Pcp2^Cre^;Slc32a1^flox/flox^* animals to demonstrate that Purkinje cell neurotransmission contributes to baseline physiological tremor. We further demonstrated that the lack of Purkinje cell neurotransmission abolishes harmaline tremor. Interestingly, the reduction in power of baseline physiological tremor of *Pcp2^Cre^;Slc32a1^flox/flox^* animals occurred at frequencies adjacent to the peak of physiological tremor, which in our mice and in humans occurs at about 10 Hz. There are a number of possibilities for how this might occur. Physiological tremor includes a wide range of frequencies that extend beyond alpha band. There are many potential contributors to physiological tremor, especially to the peak frequencies around 10 Hz, with generators of 10 Hz spikes and oscillations found throughout the motor system (*McAuley and Marsden, 2000*). At the level of the muscles, the predominant firing frequency of muscle motor units is ~10 Hz. While these spikes are not necessarily, or obligatorily, coherent with the muscle oscillations of physiological tremor, increased synchrony between motor units could contribute to a physiological tremor peak at 10 Hz (*Christakos et al., 2006*). Furthermore, oscillations as a result of spinal reflex mechanics – particularly the stretch reflex – have long been proposed as a major contributor to the 10 Hz physiological tremor peak (*Jalaleddini et al., 2017*; *Lippold, 1971*). These are in concert with the resonance of the physical mechanical properties of the body and limbs, of which the natural oscillation of a human limb is ~10 Hz (*Raethjen et al., 2000*). In the central nervous system, the gap junctions of the inferior olive contribute to a subthreshold oscillation of ~10 Hz (*Leznik and Llinás, 2005*), which becomes relevant to the pathological tremor produced by harmaline (*Park et al., 2010*). 10 Hz spikes (*Hua et al., 1998*) and local field potentials (LFPs) (*Pedrosa et al., 2014*) as well as alpha band oscillations (*Budini et al., 2014*; *Muthuraman et al., 2012*) have been detected in the thalamus in cases of tremor. Alpha band oscillations, which include 10 Hz, are also present at multiple phases of non-

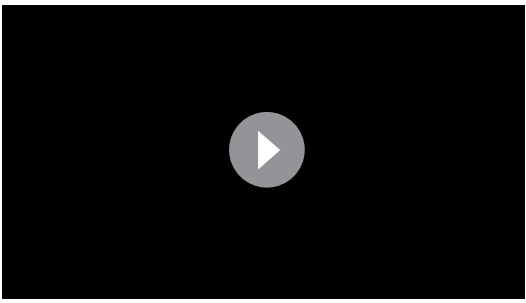

**Video 4.** Closed-loop DBS of the interposed nucleus reduces harmaline-induced tremor.
https://elifesciences.org/articles/51928#video4

**Table 5.** Precision measures, exact p-values, and replicate data relevant to *Figure 5*.

| Figure | Comparator 1 | Comparator 2 | Mean 1 | Mean 2 | SE of diff. | N 1 | N 2 | Summary | Adjusted P Value |
|---|---|---|---|---|---|---|---|---|---|
| *Figure 5i* | baseline | harmaline | 0.09208 | 1.000 | 0.03831 | 8 | 8 | **** | <0.0001 |
| | baseline | closed-loop DBS + harmaline | 0.09208 | 0.2129 | 0.07930 | 8 | 8 | ns | 0.3375 |
| | harmaline | closed-loop DBS + harmaline | 1.000 | 0.2129 | 0.06191 | 8 | 8 | **** | <0.0001 |

pathological movements (*Armstrong et al., 2018*). Furthermore, smooth movements are comprised of multiple discontinuous trajectory modifications that occur at ~10 Hz and are related to oscillations of similar frequency that emerge from the cerebello-thalamo-cortical network, potentially forming the basis of a physiological tremor peak at 10 Hz (*Gross et al., 2002*; *Schnitzler et al., 2006*; *Bye and Neilson, 2010*). Therefore, many factors may contribute to the generation and maintenance of physiological tremor at this frequency. It is possible that one of or more of these various known 10 Hz features are capable of compensating for the constitutive lack of Purkinje cell neurotransmission caused by our *Slc32a1* knockout or that eliminating Purkinje cell activity could result in subtle compensatory changes that suppress pathological tremor at this frequency (*White et al., 2014*). Additionally, it is possible that simply the lack of Purkinje cell neurotransmission does not completely offset these many varied possible contributors to the ~10 Hz physiological tremor peak. But, an effect of Purkinje cell neurotransmission on baseline physiological tremor is realized at frequencies farther away from the peak, suggesting a cerebellar contribution to physiological tremor that cannot be entirely compensated or offset (*Figure 1f and i*). Moreover, harmaline tremor, which is thought to be largely driven by the 10 Hz oscillatory capability of the inferior olive, is significantly reduced in mice lacking Purkinje cell neurotransmission (*Figure 1g–h*). This suggests a necessity of the Purkinje cells in propagating this fundamentally ~10 Hz oscillatory signal. Importantly, there appears to be a central to peripheral division in the nervous system in these 10 Hz features wherein a physiological tremor with a peak at ~10 Hz can still be detected even in deafferented cases (*Sanes, 1985*; *Marsden et al., 1967*). This suggests that features of the periphery may generate a base physiological tremor that can be added to or manipulated by central processes. Accordingly, tremor disorders may represent a shift in the weighting of a central pathophysiological signal and a peripheral physiological tremor component. In the scope of the multiple manipulations we present here: the constitutive lack of Purkinje cell neurotransmission did not create a tremorgenic central pathophysiological signal that was sufficient to disrupt or overcome the multiple other 10 Hz influences on the motor circuit, but enough change was induced to dampen frequencies away from this peak (*Figures 1f*, *2h–p* and *3f–h*). In contrast, the central tremorgenic signal caused by harmaline, which originates in the inferior olive, could be sufficient to overwhelm or add to the peripheral physiological tremor mechanisms in control animals, but the lack of Purkinje cell neurotransmission was sufficient to significantly reduce this central signal (*Figures 1g* and *3f–h*). Importantly, in the case of our optogenetically induced tremor, we show that tremor is capable of being generated at multiple frequencies, but with 10 Hz stimulation generating a peak tremor frequency (*Figure 4o*). This is perhaps due to harnessing or amplifying existing 10 Hz mechanisms in the motor system. Therefore, we show that cerebellar activity is key to the generation of a central pathophysiological tremor signal and is capable of driving a wide range of tremorgenic oscillations, which are strengthened when approaching or matching this common 10 Hz frequency. However, the broader significance and somewhat special peculiarity of the 10 Hz frequency in normal and abnormal movement remains unresolved.

We note that some studies of human cases of tremor have found Purkinje cell loss as well as Purkinje cell degeneration and abnormal morphology (*Louis, 2016*). The *Pcp2^Cre^;Slc32a1^flox/flox^* approach used here is a constitutive genetic silencing model with high efficiency that causes no known gross morphological defects in any cerebellar cell type or degeneration (*White et al., 2014*). As our model is constitutive and *Slc32a1* is removed from Purkinje cells throughout the cerebellum, it does not mimic the progressive loss of Purkinje cells or a localized insult to cerebellar circuitry that has been associated with some manifestations of tremor in humans (*Manto, 2018*; *Lai et al., 2019*). However, Purkinje cell loss is not always necessary for tremor to occur (*Morgan et al., 2017*;

*White et al., 2016a*). Therefore, because there are many forms of the disease, there could be many different potential mechanisms of tremor induction that do not involve Purkinje cell loss or degeneration (*Morgan et al., 2017*). In cases of tremor where Purkinje cell loss is reported, the loss is incomplete and leaves to question what signals the remaining Purkinje cells send and how their downstream partners in the cerebellar nuclei respond (*Axelrad et al., 2008*). In fact, cases of cerebellar stroke have been found to reduce tremor in humans (*Benito-León and Labiano-Fontcuberta, 2016*). Thus, rather than questioning the anatomic or genetic changes associated with specific forms of tremor, our experiments argue that there may be a commonality of abnormal electrical signaling patterns that result in tremor in general. In this study, we have shown that a single cell type, the Purkinje cells, have the ability to serve as a gatekeeper of tremor.

We have found that if cerebellar nuclei cells are induced into bursting patterns of activity, a tremor will result. We show this pattern of activity of the cerebellar nuclei requires intact Purkinje cell output in the case of harmaline tremor, which has been used to model tremor behavioral phenotypes for the goal of developing therapeutics (*Handforth, 2012*). Abnormal activity in the Purkinje cell to cerebellar nuclei connection is also implicated in ataxia (*Walter et al., 2006*; *Egorova et al., 2016*) and dystonia (*Calderon et al., 2011*; *Fremont et al., 2014*; *Fremont et al., 2015*; *White and Sillitoe, 2017*). This is intriguing because it raises the hypothesis that different modes of abnormal activity could contribute to ataxia, dystonia, and tremor. For example, in dystonia the cerebellar nuclei cells are induced into an irregular firing pattern with both elevated CV and CV2 (*White and Sillitoe, 2017*) while in the electrophysiological recordings during tremor that we describe here show only an elevated CV. Instead of the erratic pattern of activity seen in dystonia, we observe a regular bursting pattern of activity in tremor. It is tempting to speculate that the spike firing plus population features such as synchrony (*Sarnaik and Raman, 2018*) may distinguish these and other disease states that arise from or involve cerebellar circuitry. This idea is further supported by our data which demonstrates that multiple frequencies of cerebellar optogenetic stimulation can cause a wide range of tremor phenotypes, which in humans would equate to very different disease conditions. Our finding of optogenetically-induced tremor severity peaking at around 10 Hz matches the frequency of tremor commonly noted in humans with essential tremor (*Clark and Louis, 2018*) while the 3 Hz peak matches that frequently observed in Holmes' tremor, which has suspected cerebellar involvement (*Raina et al., 2016*). The peak at 20 Hz is intriguing as it is both a harmonic of 10 Hz and similar frequencies have been described in genetic mouse models of tremor (*Kralic et al., 2005*). Moreover, the power of the tremor at 9 Hz and 10 Hz is interesting because they both fall within the 4–12 Hz range that is key in human tremor disorders such as essential tremor (*Kuo et al., 2019*; *Kuo et al., 2018*). We therefore postulate that, whether directly driven by abnormal Purkinje cell activity or other genetic, pharmacologic, and pathologic factors that influence cerebellar circuitry, any instigator of a synchronous, regular bursting pattern of cerebellar nuclei activity could result in tremor. Together, our data provide experimental support that selective alterations in cerebellar function are capable of producing a tremor phenotype across a range of disease-relevant frequencies. The data also underscore the capacity of functional heterogeneity, notably the heterogeneity in the defects arising from a common Purkinje cell circuit, which may promote the cerebellum's involvement in multiple diseases. It is this inherent circuit flexibility that may also equip the cerebellum to contribute to a vast number of motor as well as non-motor behaviors.

Finally, our ability to acutely disrupt ongoing tremor behavior using closed-loop DBS suggests that the cerebellum itself may be an ideal target for the treatment of intractable tremor. This is consistent with the current practice of targeting DBS electrodes to regions of the thalamus that receive cerebellar input (*Miterko et al., 2019*). In this study, we chose to use high-frequency DBS for the treatment of the predominant tremor phenotype that results from harmaline administration. This is because high-frequency DBS (>100 Hz) directed to extra-cerebellar targets is typically used for the treatment of tremor disorders such as in human patient groups with intractable essential tremor, Parkinson's disease, and other disorders that can involve tremor (*Shields et al., 2011*; *Valálik et al., 2012*; *Su et al., 2018*; *Barbe et al., 2018*; *Pahwa et al., 2006*). Additionally, our previous work showed that similar high-frequency DBS directed to the cerebellar nuclei ameliorates dystonia in mice (*White and Sillitoe, 2017*). It is possible that other frequencies of stimulation directed towards the cerebellum may also be capable of reducing tremor severity, particularly when there are other motor deficits present (*Anderson et al., 2019*) and, therefore, future studies comparing the efficacy of various frequencies of cerebellar DBS are warranted. Still, the addition of tremor as another

hyperkinetic movement disorder that can be treated with cerebellar DBS provides hope that other cerebellum-associated movement disorders may be addressable with cerebellar DBS. Indeed, cerebellar DBS may be especially useful with patients who present with multiple cerebellum-related features, such as tremor with ataxia or tremor with dystonia (*Anderson et al., 2019*; *Oyama et al., 2014*). Moreover, the need for alternative targets has become apparent as some patients develop tolerance to thalamic DBS (*Favilla et al., 2012*) or exhibit cognitive and motor decline as a result of current stimulation practices (*Woods et al., 2003*; *Reich et al., 2016*). Importantly, in humans, targeting the cerebellar nuclei with neuromodulation has shown great promise after cerebellar stroke (*Teixeira et al., 2015*) and, specifically for DBS, functional improvements were reported after dentate-directed DBS in rat models of cortical stroke (*Cooperrider et al., 2014*; *Shah et al., 2017*). Dentate stimulation at low frequencies was also shown to be effective in the shaker rat model of neurodegenerative ataxia, a model with Purkinje cell loss which also exhibits tremor (*Anderson et al., 2019*). Our previous work showed strong modulation of dystonia-like behaviors with cerebellar interposed DBS (*White and Sillitoe, 2017*). However, given that there are multiple thalamic targets of the cerebellar nuclei (*Gornati et al., 2018*; *Teune et al., 2000*), it is possible that applying our closed-loop stimulation approach to the fastigial and dentate nuclei could also be effective in reducing tremor. Moreover, within the interposed nucleus itself, it is possible that stimulating the anterior versus the posterior regions could produce different impacts during tremor suppression. This hypothesis is supported by the unique projection patterns that help define the interposed regions (*Teune et al., 2000*) and since each region could influence distinct aspects of movement such as speed or direction (*Valle et al., 2010*). While our understanding of the therapeutic mechanisms of DBS remains incomplete, both the data we present in this study and our previous work suggest that the disruption of abnormal patterns of Purkinje cell neurotransmission to the cerebellar nuclei and/or correction of abnormal cerebellar nuclei activity may be instrumental to the success of cerebellar DBS (*White and Sillitoe, 2017*). That said, there are a number of possible mechanisms that could be harnessed by DBS (*Herrington et al., 2016*). One hypothesis is that the DBS pulses produce an inhibitory effect on neuronal somata, with the major impact on those neurons that are proximal to the electrode. In this scenario, the inhibitory influence of the DBS could be induced by a depolarization block, perhaps through a mechanism involving sodium channel inactivation and potassium current potentiation. However, it is also possible that DBS enhances activity at the stimulation location by exciting local axons. Whatever the mechanism, the end result is that the entrainment from the DBS overrides pathological activity. In tremor, we propose that a significant source of the abnormal behavior is due electrophysiological defects that stem from changes in the firing regularity of cerebellar nuclei neurons. During closed-loop DBS, it is possible that activity within cerebellar long-distance connections and the oscillations that drive the behavior are normalized. However, given the potential dependence of cerebellar function – especially oscillations – on neuronal synchrony (*Welsh et al., 1995*), it could be that DBS circuit modulation also involves the normalization of patterned activity in the local circuit. Based on the potential for retrograde signaling from the stimulating electrode, another possibility is that climbing fiber activity is modulated, either at the collaterals in the cerebellar nuclei or at their direct input to Purkinje cells. The climbing fiber to Purkinje cell synapse is critical for synchrony within the cerebellar modular architecture (*Schultz et al., 2009*; *Welsh, 2002*), a framework that would facilitate the communication with the cerebellar nuclei (*Person and Raman, 2012*). We show here that an abnormal pattern of activity that is transmitted from the Purkinje cells downstream to the cerebellar nuclei cells can be interrupted for recovery from tremor. In all, our data implicate a cerebellar circuit mechanism of tremor that may operate across tremor disorders and highlight the cerebellum as a potential target for tremor therapy.

## Materials and methods

### Mouse lines

All experiments were performed according to a protocol approved by the Institutional Animal Care Use Committee (IACUC) of Baylor College of Medicine. Both male and female adult mice, at least two months and less than 8 months of age, were studied. All mice were kept on a 14 hr/10 hr light/dark cycle. Purkinje cell specificity was achieved using a *Pcp2^Cre^* transgenic mouse line, also known

as *L7^{Cre}* (*Lewis et al., 2004*). Genetic removal of Purkinje cell GABA neurotransmission was achieved by crossing this line to one that expresses a knock-in floxed *Slc32a1* allele, also known as *Vgat* (*Tong et al., 2008*). Optogenetic manipulation of Purkinje cells was achieved by crossing the *Pcp2^{Cre}* line to a reporter line that expresses channelrhodopsin (ChR2) fused to enhanced yellow fluorescent protein (EYFP) behind a *floxed-stop* cassette that was targeted to the *Rosa26* locus (The Jackson Laboratory, Bar Harbor, Maine, USA strain #024109, Ai32(RCL-ChR2(H134R)/EYFP)) (*Madisen et al., 2012*). In the cerebellar system, EYFP reporter expression was only observed in Purkinje cells after performing the cross to the *Pcp2^{Cre}* mice, whereas connected structures such as the inferior olive remained negative for the reporter (*Figure 4—figure supplement 1*). During breeding, we considered the day a vaginal plug was visible as E0.5 and the day of birth as P0. We used standard genotyping protocols and primers for *Cre* and *Gfp* (to detect *Eyfp*) as described previously (*White et al., 2014*; *White and Sillitoe, 2017*) and the *Slc32a1 floxed* allele detected as originally published (*Tong et al., 2008*). See below for further details on anatomy, electrophysiology, and behavior.

## Tremor recording and analysis

Tremor was detected using at least one of three methods, including two tremor monitors and EMG. Tremor monitors used included a commercial model (San Diego Instruments, San Diego, CA, USA) as well as a custom built model that was inspired by a previously used design (*Park et al., 2010*). In the commercial model setup, mice are placed inside a clear plastic tube that is fused to a small platform with rounded legs. An accelerometer is mounted to the bottom of the platform beneath the mouse and detects the shaking of the platform caused by the mouse's tremor. The entire setup is placed inside of a sound-reducing opaque box. In the custom-built model, mice are placed into a translucent plastic box with an open top. The box is held steady in air by eight elastic cords, one end of each cord is connected to a separate corner of the box while the opposite ends are connected to vertical metal rods that form a perimeter around the box. The elastic cords that are connected to the top corners of the box are attached to the top of the nearest metal rod while the elastic cords that are connected to the bottom corners of the box are attached to the bottom of the nearest metal rod. The top cords provide upwards tension while the bottom cords provide downwards tension.

Mice were allowed to acclimate to the tremor monitor for a period ranging between 120 to 500 s before recordings of tremor were made. For both EMG and tremor monitor recordings, power spectrums of tremor were made using a fast Fourier transform (FFT) with a Hanning window. An offset was applied if the tremor waveform was not centered on 0 and the recordings were downsampled using the Spike2 software 'interpolate' channel process in order to produce frequency bins aligned to whole numbers. FFT frequency resolution was targeted to either ~0.25 Hz or ~0.5 Hz per bin. Sonogram plots of tremor were made using the Spike2 software 'sonogram' channel draw mode with a Hanning window. Power of tremor within a band was calculated by summing the power of each frequency within the band. Alpha + beta band was considered to be 8 Hz to 19.5 Hz. Gamma band was considered to be 20 Hz to 30 Hz. We report frequency ranges in order to account for natural variability in tremor signals. Alpha and beta bands were summed in order to capture the entire harmaline tremor peak, to account for the key tremor ranges and variability in peaks between animals, and to prevent splitting the power of any given interesting tremor peaks into multiple analyzed sets of data. Brown-Forsythe and Welch ANOVA tests with Dunnett's T3 multiple comparisons test were performed to determine whether band power was significantly different between conditions. For normalized peak comparison, an RM one-way ANOVA with Geisser-Greenhouse correction and Tukey's multiple comparison's test were used. A minimum of 25 s and up to ~120 s of the tremor recordings were analyzed for each animal in each time period.

For analysis of sex differences within test conditions, statistical significance was determined using multiple t-tests with the Holm-Sidak method with alpha = 0.05. Each test condition (genotype + whether harmaline had been administered) was analyzed individually, without assuming a consistent standard deviation. Reported P values are adjusted for multiple comparisons.

Number of animals tested is represented by 'N'. 'Control' refers to *Slc32a1^{flox/flox}* animals while 'mutant' refers to *Pcp2^{Cre};Slc32a1^{flox/flox}* animals. P value > 0.05 = ns, $\leq$0.05 = *, $\leq$0.01 = **, $\leq$0.001 = ***, <0.0001 = ****.

## Administration of harmaline

Adult mice were administered 30 mg/kg harmaline (Sigma-Aldrich, St. Louis, MO, USA; #H1392) via intraperitoneal injection (IP). Harmaline tremor consistently developed between 5 to 15 min after injection. Mice were administered only one dose of harmaline and were sacrificed within 4 hr of the injection.

## Immunohistochemistry

Perfusion and tissue fixation were performed as previously described (*Sillitoe et al., 2008*). In short, mice were anesthetized with Avertin (2, 2, 2-Tribromoethanol, Sigma-Aldrich, St. Louis, MO, USA; #T48402) via intraperitoneal injection. Once mice were deeply anesthetized, a whole-body perfusion was performed first with ice-chilled 0.1M phosphate-buffered saline (PBS; pH 7.4), then with ice-chilled 4% paraformaldehyde (4% PFA) diluted in PBS. The brain was then dissected out and placed in 4% PFA for 24 to 48 hr for post-fixation at 4℃. Cryoprotection was then performed by placing the tissue in stepwise sucrose dilutions, first in 15% sucrose in PBS followed by 30% sucrose in PBS. Brains were stored at 4℃ during stepwise sucrose incubation steps. After cryoprotection, the tissue was embedded in Tissue-Tek O.C.T. Compound (Sakura, Torrance, CA, USA) and frozen at −80℃. Tissue sections were then cut on a cryostat at 40 µm thickness and stored free-floating in PBS at 4℃.

Immunohistochemistry procedures have been described previously (*White et al., 2014*; *Sillitoe et al., 2003*; *White and Sillitoe, 2013*; *Sillitoe et al., 2010*). C-Fos staining was performed using rabbit polyclonal anti-c-Fos (Santa Cruz Biotechnology, Dallas, TX, USA; #sc-52). Signal was amplified using a Vectastain ABC kit (Vector Laboratories, Burlingame, CA, USA; #PK-6100) and followed with biotinylated goat anti-rabbit antibodies (Vector Laboratories, Burlingame, CA, USA; #BA-1000). Finally a 3, 3′-diaminobenzidine (DAB; Sigma-Aldrich, St Louis, MO, USA; #D5905-50TAB) reaction was used to reveal the antibody binding. After staining, sections were mounted on electrostatically coated slides with mounting medium (Vector Laboratories, Burlingame, CA, USA; #H-1200, Electron Microscopy Sciences, Hatfield, PA, USA; #17985–11 or Thermo Fisher Scientific, Waltham, MA, USA; #8312–4) and imaged. Triple fluorescent staining was completed using rabbit polyclonal anti-carbonic anhydrase VIII (Car8) to visualize Purkinje cells (CAVIII, Santa Cruz Biotechnology, #sc-67330), chicken anti-GFP to visualize ChR2 (Abcam, Cambridge, UK, #AB13970), and NeuroTrace fluorescent Nissl 530/615 to visualize neurons (Molecular Probes Inc, Eugene, OR, USA, #N21482). Secondary antibodies included Alexa-488 and −647 secondary goat anti-mouse and anti-rabbit antibodies (Molecular Probes Inc, Eugene, OR, USA).

Nissl staining was performed by first mounting the tissue sections on gelatin coated slides and allowing them to dry on the slides overnight. Mounted sections were then immersed in 100% xylene two times for 5 min and then put through a rehydration series which consisted of 3 immersions in 100% ethanol followed by 95% ethanol, 70% ethanol, and tap water, for two minutes per step. Sections were then immersed in cresyl violet solution for ~10 min or until the stain was sufficiently dark. The sections were then dehydrated using the reversed order of the rehydration series followed by xylene, with 30 s to 1 min per step. Cytoseal XYL mounting media (Thermo Scientific, Waltham, MA, USA, #22-050-262) and a coverslip was then immediately placed on the slides.

## Imaging of immunostained tissue sections

Photomicrographs of stained tissue sections were captured using either a Zeiss Axio Imager.M2 microscope equipped with Zeiss AxioCam MRm and MRc5 cameras (Zeiss, Oberkochen, Germany) or a Leica DM4000 B LED microscope equipped with Leica DFC365 FX and Leica DMC 2900 cameras (Leica Microsystems Inc, Wetzlar, Germany). Zeiss Zen software was used for image acquisition from the Zeiss microscope. Leica Application Suite X (LAS X) software was used for image acquisition from the Leica microscope. Images were corrected for brightness and contrast using Adobe Photoshop CS5 (Adobe Systems, San Jose, CA, USA) for figure preparation. Schematics were made in Adobe Illustrator CC.

## Quantification of c-Fos

Photomicrographs of cerebellar nuclei (control N = 4, mutant N = 4) were imported into Fiji program (imageJ) (*Schneider et al., 2012*; *Schindelin et al., 2015*; *Schindelin et al., 2012*). The scale of the images was set in the program and images were converted to 16-bit. A threshold was set on the

image so as to detect only c-Fos expression. A watershed process was applied to the detected puncta to separate any 'clumped' puncta into individual punctum. The 'analyze particles' function was used to both count and quantify the size of the detected puncta within hand-drawn ROIs set around each of the cerebellar nuclei. Puncta density was defined as the number of puncta within the area of the ROI. Puncta coverage was defined as the percent of the area of the ROI that was covered by the area of detected puncta. 2-way ANOVAs with the Sidak correction for multiple comparisons were used to analyze cerebellar nuclei data sets. Unpaired two-tailed t tests were used to analyze inferior olive data sets. Number of animals quantified is represented by 'N'. Number of photomicrographs is represented by 'n'.

## Surgery

We have previously described our general surgical techniques in detail (*White et al., 2016b*). In brief, for all surgical techniques used in these studies, mice were given preemptive analgesics (buprenorphine, 0.6 mg/kg subcutaneous (SC), and meloxicam, 4 mg/kg SC) with continued applications provided as part of the post-operative procedures. Anesthesia was induced with 3% isoflurane gas and maintained during surgery at 2% isoflurane gas. All surgeries were performed on a stereotaxic platform (David Kopf Instruments, Tujenga, CA, USA) with sterile surgery techniques. All animals were allowed to recover for at least three days to a maximum of one week after surgery. The following surgical techniques were either employed as individual experiments or combined depending on the requirements of the experiment:

## Awake head-fixed neural recordings

During surgeries for awake neural recording experiments, the dorsal aspect of the skull was exposed and a circular craniotomy of about 2 mm in diameter was performed dorsal to the interposed nucleus (6.4 mm posterior and ±1.3 mm lateral to Bregma.) A custom-built 3D-printed chamber was placed around the craniotomy and filled with antibiotic ointment. A custom headplate used to stabilize the mouse's head during recordings was affixed over Bregma, and a skull screw was secured into an unused region of skull. All implanted items were secured using C and B Metabond Adhesive Luting Cement (Parkell, Edgewood, NY, USA) followed by a layer of dental cement (A-M Systems, Sequim, WA, USA; dental cement powder #525000 and solvent #526000) to completely enclose the area.

## Optical fiber implantation

Optical fiber implant surgeries began as described above, however instead of performing a single large craniotomy, two small craniotomies were performed bilaterally and dorsal to the interposed nucleus (6.4 mm posterior and ±1.3 mm lateral to Bregma) through which two optical fibers (Thorlabs, Newton, NJ, USA; #FT200UMT) were lowered just into the region of the interposed nucleus (2.5–3.0 mm ventral from the surface of the brain). Optical fibers had been previously glued into ceramic ferrules (Thorlabs, Newton, NJ, USA; #CFLC230-10), polished (Thorlabs, Newton, NJ, USA; #LF5P, #LF3P, #LF1P, #LF03P, #LFCF), epoxied to each other at a set distance, and placed inside ceramic mating sleeves (Thorlabs, Newton, NJ, USA; #ADAL 1–5) prior to implantation. The fibers, a skull screw, and 1 or two metal rods used for holding the mouse's head stable while connecting the optical fiber patch cables were affixed to the skull using C and B Metabond Adhesive Luting Cement followed by dental cement.

## DBS electrode implantation

DBS electrode implant surgeries were performed exactly as the optical fiber implantation surgeries. Except, instead of optical fibers, custom 50 mm twisted bipolar Tungsten DBS electrodes were used (PlasticsOne, Roanoke, VA, USA; #8IMS303T3B01).

## EMG implantation

Surgeries for EMG electrodes required only exposing the skull before an incision was made into the left hind limb. Handmade silver wire electrodes (A-M Systems, Sequim, WA, USA; #785500) were then implanted into the gastrocnemius of the left hind limb. The electrode wire was fed under the skin from the hind limb to the skull. An additional ground electrode was implanted under the skin of

the neck. These wires were soldered to a connector for a detachable preamplifier (Pinnacle Technology, Inc, Lawrence, KS, USA; #8406). The connector, a skull screw, and 1 or two metal rods used for holding the mouse's head stable while connecting the preamplifier were affixed to the skull using C and B Metabond Adhesive Luting Cement followed by dental cement.

## In vivo electrophysiology

Single-unit extracellular recordings were performed as previously described (*White et al., 2016a*; *White and Sillitoe, 2017*; *Arancillo et al., 2015*). In the harmaline experiments, mice were awake and head-fixed to a frame while standing on a foam wheel which reduced the force they were able to apply to the headplate. Before recordings, mice were trained to become accustomed to being in the recording setup and head-fixed, which typically involved three 30 min sessions. At the time of the recordings, the chamber around the craniotomy was emptied of antibiotic ointment and refilled with 0.9% w/v NaCl solution. Electrodes had an impedance of 4–13 MΩ and were made of either tungsten (Thomas Recording, Giessen, Germany) or glass (Harvard Apparatus, Cambridge, MA, USA; #30–0057), which had been pulled at the time of recording (Sutter Instrument, Novato, CA, USA; #P-1000) and filled with 0.9% w/v NaCl solution. For recordings of single cells during optogenetic stimulation, mice were anesthetized with ketamine 80 mg/kg admixed with xylazine (16 mg/kg). Glass electrodes as described above were used, with the addition of an optical fiber that passed through the center of the glass electrode. The end of the optical fiber was as close to the tip of the glass electrode as possible without occluding or breaking the tip. Electrodes were connected to a preamplifier headstage (NPI Electronic Instruments, Tamm, Germany). The headstage was attached to a motorized micromanipulator (MP-225; Sutter Instrument Co., Novato, CA, USA). Headstage output was amplified and bandpass filtered at 0.3–13 kHz (ELC-03XS amplifier, NPI Electronic Instruments, Tamm, Germany) before being digitized (CED Power 1401, CED, Cambridge, UK), recorded, and analyzed using Spike2 software (CED, Cambridge, UK). Electrical activity was additionally monitored aurally using an audio monitor (AM10, Grass Technologies, West Warwick, RI, USA) connected to the output of the amplifier.

Purkinje cells were identified by their firing rate and the presence of both complex and simple spike activity. The chances of finding Purkinje cells based on location was also monitored as only Purkinje cells that were isolated superficial to the nuclei were included in this study. Purkinje cells were recorded in lobules IV, V, and VI of the vermis and in the adjacent paravermis regions. Accordingly, the cerebellar nuclei cells were identified by their relatively deep location within the cerebellum, approximately 2.5–3 mm deep, and their firing rate. The surface of the tissue was determined based on the significant reduction of electrical noise that occurs when the electrode, initially suspended in air, touches the tissue. However, a thin layer of antibiotic ointment or saline sometimes remained on top of the tissue. Therefore, to account for this volume, and any minor variations in the angle of electrode penetration, the maximum allowed depth of the recording electrode from this point was 4 mm, at which point we could be certain that we had exhausted the possibility of likely finding cerebellar nuclei neurons and had therefore fully traversed the nuclei territory. During recording sessions, the experimenter monitored the sound of the electrical activity to determine whether tissue membranes were being breached, white matter tracts were being traversed, or whether sound quality deviated significantly from traditional cerebellar recordings. Using these criteria we have consistently been able to target the cerebellar nuclei (*White et al., 2014*). Cerebellar nuclei cells were recorded in the interposed nucleus. These locations were chosen because, historically, it has been understood that Purkinje cells in regions of the cerebellar cortex that project to the fastigial and interposed nuclei are more likely to be affected by harmaline (*Bernard et al., 1984*). Additionally, our studies of cFos activation (*Figure 1j–q*; *Figure 1—figure supplement 3*) suggested the greatest magnitude of effect of Purkinje cell activity on abnormal activation of cerebellar nuclei cells in our mouse models occurred in the interposed nucleus. Stable, clear, and continuous single-unit recordings of at least 30 s duration were included in the analysis. Firing properties were analyzed using Spike2 (CED, Cambridge, UK), Microsoft Excel (Microsoft, Redmond, WA, USA), custom MATLAB code (MathWorks, Natick, MA, USA), and GraphPad Prism (GraphPad Software, La Jolla, CA, USA) software. Outlier testing was performed on extracellular electrophysiology recording data using the GraphPad PRISM ROUT method with Q = 0.1%. Any outlier cell identified using this method was excluded. The number of animals tested is represented by 'N' while the number of cells recorded is represented by 'n'.

'Control' refers to $Slc32a1^{flox/flox}$ animals while 'mutant' refers to $Pcp2^{Cre};Slc32a1^{flox/flox}$ animals. P value > 0.05 = ns, $\leq$0.05 = *, $\leq$0.01 = **, $\leq$0.001 = ***, <0.0001 = ****.

## Optogenetic stimulation

Optical fibers were either implanted into the region of the interposed nucleus (see above) or held close to the tip of a recording electrode via an optopatcher (ALA Scientific Instruments Inc, Farmingdale, NY, USA) or a custom-built electrode setup. Stimulation patterns were programmed and recorded using Spike2 software and delivered using a CED Power1401 data acquisition interface (CED, Cambridge, UK) to control a 465 nm LED (ALA Scientific Instruments Inc, Farmingdale, NY, USA). Maximum LED power at the end of the implanted fiber was measured to be ~3.6 mW and stimulation consisted of sinusoidal pulses of light, from light off to maximum brightness to light off. 3.6 mW was sufficient to result in tremor for all animals tested and often resulted in a very severe tremor (*Figure 4—figure supplement 3*). This light power was calculated to be capable of driving neurons at a distance of about 0.8 mm away from the fiber tip, using the threshold of 1 mW/mm$^2$ (*Deisseroth, 2012*; *Yizhar et al., 2011*). 0.43 mW was the minimum light power that could drive an increase in tremor detectable by our custom-built tremor monitor (*Figure 4—figure supplement 3*). This light power was calculated to be capable of driving neurons at a distance of about 0.3 mm away from the fiber tip, using the threshold of 1 mW/mm$^2$ (*Deisseroth, 2012*; *Yizhar et al., 2011*). It is noted that, at the maximum light intensity, it is likely that neurons in both the anterior and posterior interposed nuclei are affected and based on the spread of light we cannot exclude the possibility that some neurons in the fastigial and perhaps even the dentate nuclei are also impacted by the light. Importantly, tremor is able to be driven at minimal light intensities at which the majority of recruited cells would be from the interposed (*Figure 4—figure supplement 2*).

The ability to elicit tremor was tested at light stimulation frequencies from 1 to 20 Hz. Light pulses gradually increased to maximum power and decreased until the light was off sinusoidally. All pulse patterns had equal duration of light-off and light-on times. Light on times were flanked by light off times. The following are the pulse durations used for each stimulation frequency: 1 Hz = 500 ms, 2 Hz = 250 ms, 3 Hz = 166.65 ms, 4 Hz = 125 ms, 5 Hz = 100 ms, 6 Hz = 83.35 ms, 7 Hz = 71.45 ms, 8 Hz = 62.5 ms, 9 Hz = 55.55 ms, 10 Hz = 50 ms, 11 Hz = 45.45 ms, 12 Hz = 41.67 ms, 13 Hz = 38.46 ms, 14 Hz = 35.71 ms, 15 Hz = 33.34 ms, 16 Hz = 31.25 ms, 17 Hz = 29.41 ms, 18 Hz = 27.78 ms, 19 Hz = 26.31 ms, 20 Hz = 25 ms. Significance of tremor frequencies over baseline were calculated using multiple t-tests with the Holm-Sidak method with alpha = 0.05. Each test condition (frequency of stimulation) was analyzed individually, without assuming a consistent standard deviation. Reported P values are adjusted for multiple comparisons.

## Closed-Loop DBS

Closed-loop DBS programs were written as custom Spike2 scripts and configuration files (CED, Cambridge, UK). To stimulate in the condition of tremor, we wrote scripts that ran in the background of our recordings and functioned in concert with the configuration paradigm that acted as the stimulation generator. The scripts were designed to detect and trigger DBS based on the presence of an enhanced tremor phenotype.

To detect tremor behavior, tremor monitor activity between 0 and 25 Hz was used to prompt either the start or stop DBS stimulation depending on whether this abnormal activity was present. First our script created an analysis channel in our recordings that continuously performed a power analysis of the 0 to 25 Hz frequency range of our raw tremor monitor signal using a Fast Fourier Transform (FFT) with a resolution of 0.6104 Hz, size of 8192 points, and sampled over a period of 1.6384 s. The script sampled the output of this analysis as close to real time as possible to determine whether the power of the band had surpassed a threshold that had been set to the individual mouse's peak tremor power from the baseline recording. If the threshold had been surpassed, a start signal was generated.

The start signal prompted a simultaneously-running Spike2 configuration paradigm to initiate DBS. The program initiated a looping pulse train of 5 V pulses of 20 μs duration to be output from a Power1401 (CED, Cambridge, UK). The pulses occurred with an interval of 8 ms and began 1 ms after the start signal was received. This produced stimulation at 125 Hz. The pulses were sent to a Master-8 stimulator (A.M.P.I., Jerusalem, Israel) that allowed precision timing in the production of

60 µs square biphasic pulses. These pulses were then output to external stimulus isolators (ISO-Flex, A.M.P.I., Jerusalem, Israel) that set the DBS current to 30 µA. The pulse train lasted at minimum 36 ms. Within the final 8 ms gap between pulses of the loop, the program would check for the start signal. If the start signal was still present (i.e. tremor was ongoing), the pulse train would continue for another 36 ms. If the start signal was no longer present (i.e. tremor had returned to below threshold levels), DBS stimulation would halt until the start signal was generated again. Post hoc analysis of tremor was performed using Spike2 software and custom Matlab scripts (MathWorks, Natick, MA, USA).

## Acknowledgements

This work was supported by Baylor College of Medicine, Texas Children's Hospital, the National Institute of Neurological Disorders and Stroke (AMB: F31NS101891, JJW: F31NS092264, RVS: R01NS089664 and R01NS100874), the Eunice Kennedy Shriver National Institute of Child Health and Human Development (U54HD083092), a BCM IDDRC Project Development Award, the Hamill Foundation, the Mrs. Clifford Elder White Graham Endowed Research Fund, and the Bachmann-Strauss Dystonia and Parkinson Foundation, Inc. The content is solely the responsibility of the authors and does not necessarily represent the official views of the National Center for Research Resources or the National Institutes of Health.

## Additional information

### Funding

| Funder | Grant reference number | Author |
|---|---|---|
| National Institute of Neurological Disorders and Stroke | F31NS101891 | Amanda M Brown |
| National Institute of Neurological Disorders and Stroke | F31NS092264 | Joshua J White |
| National Institute of Neurological Disorders and Stroke | R01NS089664 | Roy V Sillitoe |
| National Institute of Neurological Disorders and Stroke | R01NS100874 | Roy V Sillitoe |
| Eunice Kennedy Shriver National Institute of Child Health and Human Development | U54HD083092 | Roy V Sillitoe |
| Baylor College of Medicine | IDDRC Project Development Award | Roy V Sillitoe |
| Texas Children's Hospital | | Roy V Sillitoe |
| Hamill Foundation | | Roy V Sillitoe |
| Baylor College of Medicine | Mrs. Clifford Elder White Graham Endowed Research Fund | Roy V Sillitoe |
| Bachmann-Strauss Dystonia and Parkinson Foundation | | Roy V Sillitoe |

The funders had no role in study design, data collection and interpretation, or the decision to submit the work for publication.

### Author contributions

Amanda M Brown, Joshua J White, Roy V Sillitoe, Conceptualization, Resources, Data curation, Software, Formal analysis, Supervision, Funding acquisition, Validation, Investigation, Visualization, Methodology, Project administration; Meike E van der Heijden, Conceptualization, Resources, Data curation, Software, Formal analysis, Supervision, Validation, Methodology; Joy Zhou, Conceptualization, Resources, Data curation, Software, Formal analysis, Validation, Methodology; Tao Lin, Resources, Supervision, Validation, Investigation, Methodology, Project administration

## Author ORCIDs

Amanda M Brown [ID] https://orcid.org/0000-0002-1484-8972
Meike E van der Heijden [ID] http://orcid.org/0000-0003-0801-8806
Joy Zhou [ID] http://orcid.org/0000-0003-1731-8800
Roy V Sillitoe [ID] https://orcid.org/0000-0002-6177-6190

## Ethics

Animal experimentation: Mice were housed in an AAALAS-certified animal facility. All procedures to maintain and use these mice were approved by the Institutional Animal Care and Use Committee for Baylor College of Medicine (Animal protocol number AN-5996).

## Decision letter and Author response

Decision letter https://doi.org/10.7554/eLife.51928.sa1
Author response https://doi.org/10.7554/eLife.51928.sa2

## Additional files

### Supplementary files

• Transparent reporting form

### Data availability

All data is available in the main text, supplementary materials, or supporting files.

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
