## [Decision Letter]

**Acceptance summary:**

This very interesting study uses genetic, pharmacologic and optogenetic manipulations to determine the direct role of the cerebellum, and in particular Purkinje cells and deep cerebellar nucleus neurons in the generation of tremor. Further, the authors demonstrate a proof-of-principle for the use of Deep Brain Stimulation (DBS) in the cerebellar nuclei as a therapy for tremor symptoms.

**Decision letter after peer review:**

Thank you for submitting your article "Cerebellar Purkinje cell microcircuits are essential for tremor" for consideration by *eLife*. Your article has been reviewed by Richard Ivry as the Senior Editor, a Reviewing Editor, and three reviewers. The following individual involved in review of your submission has agreed to reveal their identity: Freek E Hoebeek (Reviewer #1).

The reviewers have discussed the reviews with one another and the Reviewing Editor has drafted this decision to help you prepare a revised submission.

The reviewers agree that the results are interesting and appropriate for *eLife*. However, important issues were raised, particularly about the localization of the recording and DBS electrodes, cell-type specificity, and the generality of results across different kinds of tremor. These concerns are summarized here with suggestions for how they could be addressed.

Essential revisions:

1) Throughout the manuscript, there is missing information about localization of the manipulations.

a) Which and how many IP/ CN neurons need to be entrained to induce tremor? Please report the anatomical location of the recordings and stimulation sites. In addition, the potential influence of stimulating the anterior and/or posterior interposed should be discussed.

b) Which Purkinje cells were recorded? Where were they located, in which lobule(s)?

c) Please also be sure to comment on the motivation for targeting these areas. Is there a rationale linked with an 'area of tremor'? How were locations for recording and stimulation chosen?

2) Additional analyses, data collection.

a) Please estimate the required extent of stimulation in the cerebellar nuclei for inducing tremor. This could be done by repeating the experiment where the example trace has already been shown in Figure 4J, varying the stimulus/light intensity and calculating off-line how far from the optrode neurons have been stimulated.

b) Please quantify the c-fos experiment. For instance, in the images shown now, the cfos expression in the lateral CN is lower than in the medial and interposed CN. Is that consistent? Another question arising from the images is that the CN in Figure 1P is not empty, indicating that several cells are still showing increased activity levels compared to saline injections. Is this consistent? Might this be related to the climbing fiber collaterals that innervate the CN?

3) Frequency of tremor and of DBS.

a) Multiple reviewers commented on the broad range of tremor frequencies reported. Please quantify the power spectrum of tremor frequencies.

b) While the deep brain stimulation studies are acceptable as proof-of-concept, the effectiveness of the specific frequency of deep brain stimulation for cerebellar tremor should be examined. A recent example of this technique is in Anderson et al., (2019), demonstrating that 30 Hz stimulation improved tremor and ataxia in a rat model. It is important to cite this paper.

4) Cell-type specificity.

a) Please confirm the specificity of L7-Cre driven expression to Purkinje cells and examine possible expression in the inferior olive. If specificity cannot be demonstrated, the possibility of cre-based excision in the inferior olive should be included.

b) Recording from Purkinje neurons with DBS on may help determine whether the effect of DBS is on cerebellar output versus cerebellar climbing fibers.

c) It may be necessary to include the possibility that the mechanism is through DBS mediated inhibition of climbing fiber input, if this cannot be confirmed experimentally.

5) Finally, the generality of these results for all kinds of tremor has not been established. The title and conclusions of the paper should be toned down. One reviewer noted that It is not entirely clear which cerebellar "microcircuits" are responsible for tremor. Would a more appropriate title and conclusions, based on the data provided, be that Purkinje neurons are essential for harmaline tremor, and that high-frequency closed-loop DBS to the cerebellum improves this tremor?

---

## [Author Response]

Essential revisions:1) Throughout the manuscript, there is missing information about localization of the manipulations.a) Which and how many IP/ CN neurons need to be entrained to induce tremor? Please report the anatomical location of the recordings and stimulation sites. In addition, the potential influence of stimulating the anterior and/or posterior interposed should be discussed.b) Which Purkinje cells were recorded? Where were they located, in which lobule(s)?c) Please also be sure to comment on the motivation for targeting these areas. Is there a rationale linked with an 'area of tremor'? How were locations for recording and stimulation chosen?

Thank you for pointing these issues out to us. In the respective sections of the Results section, we have now provided the rationale for targeting the interposed cerebellar nuclei, and also for collecting Purkinje cell activity data from lobules IV, V, and VI of the vermis and from the adjacent paravermal regions. We have now specified the exact coordinates used for recording and stimulating the interposed nuclei in the methods.

2) Additional analyses, data collection.a) Please estimate the required extent of stimulation in the cerebellar nuclei for inducing tremor. This could be done by repeating the experiment where the example trace has already been shown in Figure 4J, varying the stimulus/light intensity and calculating off-line how far from the optrode neurons have been stimulated.

These are excellent suggestions. We have tested the extent of stimulation required to generate tremor by systematically stimulating mice at a range of frequencies while recording their responses with our tremor monitor. These data are now provided in a new figure, Figure 4 - figure supplement 3. As suggested, we have also used two different methods to estimate how far from the optrode neurons are likely stimulated. First, we dissected the cerebellum in such a way that the optic fibers were left intact, and then we turned on the light to show an in situ proxy for what stimulation would look like in vivo. Second, as recommended, we also provide a calculation for how far the light might travel in the cerebellum based on the general properties of the fibers and the light, using a method that was developed as an online resource. These data can now be found in the new Figure 4 - figure supplement 2.

b) Please quantify the c-fos experiment. For instance, in the images shown now, the cfos expression in the lateral CN is lower than in the medial and interposed CN. Is that consistent? Another question arising from the images is that the CN in Figure 1P is not empty, indicating that several cells are still showing increased activity levels compared to saline injections. Is this consistent? Might this be related to the climbing fiber collaterals that innervate the CN?

We have provided a full quantification of the c-Fos expression in the control and mutant, treated with harmaline. Indeed, the fastigial and interposed have higher c-Fos expression compared to the dentate. You are absolutely correct, there is a strong possibility that climbing fiber collaterals are able to induce some level of c-Fos. We have added this discussion to the text. These data can now be found in the new Figure 1 - figure supplement 3.

3) Frequency of tremor and of DBS.a) Multiple reviewers commented on the broad range of tremor frequencies reported. Please quantify the power spectrum of tremor frequencies.

Wonderful suggestion, thank you. We have now quantified all frequencies analyzed. Please note that while tremor was visible and clear at all frequencies, the variability at specific frequencies, for example at 15 Hz, indicated no significant increase in tremor compared to baseline. We suspect that this variability can arise due to features of the experimental set up such as the use of EMG to record the tremor, which can be sensitive to noise during motion.

b) While the deep brain stimulation studies are acceptable as proof-of-concept, the effectiveness of the specific frequency of deep brain stimulation for cerebellar tremor should be examined. A recent example of this technique is in Anderson et al., (2019), demonstrating that 30 Hz stimulation improved tremor and ataxia in a rat model. It is important to cite this paper.

Thank you for spotting this unintended omission. Anderson et al., indeed provide a compelling case for using different DBS parameters (lower, in their case), which was especially effective in their shaker rat model which exhibits a tremor that slowly dampens, an ataxia that progressively gets worse, and Purkinje cell degeneration and eventual cell loss. We have now discussed the Anderson et al. paper and have also further discussed our rationale for choosing high frequency DBS (here, defined as greater than 100 Hz), and discussed the human studies that effectively use high frequency stimulation to treat motor diseases that involve tremor.

4) Cell-type specificity.a) Please confirm the specificity of L7-Cre driven expression to Purkinje cells and examine possible expression in the inferior olive. If specificity cannot be demonstrated, the possibility of cre-based excision in the inferior olive should be included.

Good point. We have confirmed that the *L7^Cre^* allele recombines and drives reporter expression in Purkinje cells, but not the inferior olive. We have provided the description for this confirmation in the Material and Methods section. In addition, these data can now be found in the new Figure 4 - figure supplement 1.

b) Recording from Purkinje neurons with DBS on may help determine whether the effect of DBS is on cerebellar output versus cerebellar climbing fibers.c) It may be necessary to include the possibility that the mechanism is through DBS mediated inhibition of climbing fiber input, if this cannot be confirmed experimentally.

This is an interesting suggestion, and indeed this is experimentally challenging to test with our current setup since combined stimulation + recording produces a high level of noise in our single unit approach. Still, we agree with the possibility that climbing fibers could be involved in the DBS response. We have discussed this possibility that climbing fiber may play a role, and also provided additional discussion about how the DBS might be impacting cerebellar circuit physiology in general. Please see the revised Discussion section of the manuscript.

5) Finally, the generality of these results for all kinds of tremor has not been established. The title and conclusions of the paper should be toned down. One reviewer noted that It is not entirely clear which cerebellar "microcircuits" are responsible for tremor. Would a more appropriate title and conclusions, based on the data provided, be that Purkinje neurons are essential for harmaline tremor, and that high-frequency closed-loop DBS to the cerebellum improves this tremor?

We appreciate this suggestion. The new title of the manuscript is "Purkinje cell misfiring generates high-amplitude action tremors that are corrected by cerebellar deep brain stimulation".

Please note that we chose not to exclusively focus on harmaline in the title due to our data covering the optogenetic induced tremors. Still, we think the new title captures the sentiment of the reviewer's suggestion.